# Quantifying Elicitation of Latent Capabilities in Language Models

**Elizabeth Donoway**[1,2][*]       **Hailey Joren**[1]        **Henry Sleight**[3]        **Arushi Somani**[1]

**John Schulman**[4]        **Julian Michael**[5]        **Michael R DeWeese**[2]        **Ethan Perez**[1]

**Fabien Roger**[1]                                **Jan Leike**[1]

[†][1]Anthropic, [2]University of California, Berkeley, [3]Constellation, [4]Thinking Machines, [5]Scale AI

## Abstract

Large language models often possess latent capabilities that lie dormant unless explicitly elicited, or surfaced, through fine-tuning or prompt engineering. Predicting, assessing, and understanding these latent capabilities pose significant challenges in the development of effective, safe AI systems. In this work, we recast elicitation as an information-constrained fine-tuning problem and empirically characterize upper bounds on the minimal number of parameters needed to achieve specific task performances. We find that training as few as 10–100 randomly chosen parameters—several orders of magnitude fewer than state-of-the-art parameter-efficient methods—can recover up to 50% of the performance gap between pretrained-only and full fine-tuned models, and 1,000s to 10,000s of parameters can recover 95% of this performance gap. We show that a logistic curve fits the relationship between the number of trained parameters and model performance gap recovery. This scaling generalizes across task formats and domains, as well as model sizes and families, extending to reasoning models and remaining robust to increases in inference compute. To help explain this behavior, we consider a simplified picture of elicitation via fine-tuning where each trainable parameter serves as an encoding mechanism for accessing task-specific knowledge. We observe a relationship between the number of trained parameters and how efficiently relevant model capabilities can be accessed and elicited, offering a potential route to distinguish elicitation from teaching.

## 1    Introduction

Many recent efforts in advancing the capability frontier of large language models (LLMs) focus on post-training methods, such as fine-tuning, reinforcement learning with human feedback, or prompt engineering. In many cases, these methods easily achieve significant performance gains through small adjustments to a model, suggesting pre-existing capabilities that need only be unlocked rather than taught. As a result, evaluations intended to benchmark maximal performance can underestimate a model's true capabilities [1], leading to an "elicitation gap" between its actual ceiling performance and the capabilities it ordinarily demonstrates. This poses challenges in accurately predicting the

---

[*]Correspondence to `donoway@berkeley.edu`
[†]Code repository: https://github.com/edonoway/quantifying-elicitation-neurips25

39th Conference on Neural Information Processing Systems (NeurIPS 2025).

model's behavior once released, with broader safety implications in ensuring deployed models aren't capable of causing harm.

Prior work that aims to extrapolate LLM capabilities has focused on the impact of scale—compute, data, or model size—on the improvement of model capabilities [2]. We propose a complementary framework that takes these as constant and instead varies the number of trainable parameters available during fine-tuning. Since all information contained in a model is stored in its parameters, we use this trainable parameter count as a direct proxy for the information budget available during fine-tuning. We then reconsider elicitation in terms of the minimal amount of information a model requires to demonstrate a capability, and we propose this quantity as an indicator of whether the capability had already existed within the model prior to fine-tuning.

We find that fine-tuning only a tiny fraction of a model's parameters often suffices to recover most of its latent capabilities, even when those parameters are selected completely at random (Figure 1). Training as few as ~$10\text{-}10^5$ randomly chosen, low-rank adapter weights can close 50-95% of the performance gap between zero-shot and fully fine-tuned LLMs across multiple choice and natural language generation tasks that span a variety of domains, including long-context chain-of-thought reasoning. We identify Pareto frontiers of performance versus parameter count when models are elicited, which assume no prior information about model internals. These frontiers reveal a logistic relationship between elicited performance and the logarithm of the number of trainable parameters (logistic in log-p): steep initial improvements that asymptotically plateau approaching the model's full parameter count. We find that this behavior generalizes across task formats (multiple choice and natural language generation), subject domains, and model sizes and families, including reasoning models.

Our approach may help distinguish elicitation, in which latent capabilities are surfaced, from teaching, wherein the model learns a new skill. We validate our approach by computing Pareto frontier curves on identical randomly initialized models, where our approach finds (as expected) that recovering even 50% of the performance when teaching complex capabilities from scratch requires several orders of magnitude more parameters than elicitation.

Our main contributions are:

1. **Elicitation frontiers.** We empirically characterize how elicited performance scales with the number of randomly selected trainable parameters across ten benchmark tasks, four model sizes, and two model families (Llama3 and Qwen2.5) and find that training only ~$10^1\text{-}10^5$ parameters can achieve 50-95% of the performance achieved by fine-tuning all model weights.
2. **Information-theoretic interpretation of elicitation.** We present a picture of elicitation built upon Rissanen's Minimum Description Length (MDL) principle [3] to quantify how much must be specified to a model for a latent capability to manifest, if one exists.
3. **Distinguishing elicitation from teaching.** We observe qualitative differences in the information models use to represent tasks when relevant capabilities are pre-existing versus absent, proposing that elicitation can be interpreted as a model's ability to develop short descriptions of how to effectively utilize latent skills.

## 2 Related Work

**Parameter-efficient fine-tuning.**   Parameter-efficient fine-tuning (PEFT) methods that freeze most of a model's weights while training only a small auxiliary module have become standard practice for adapting large language models to downstream tasks. Adapters [4] insert lightweight MLP blocks, while prefix-tuning [5] prepends trainable vectors, and LoRA [6] injects low-rank adapters. Despite using significantly fewer parameters than full fine-tuning, LoRA often nearly matches its performance, indicating that adaptation information can be highly compact [7]. Recent approaches like DoRA [8], PiSSA [9], BitFit [10], and sparse fine-tuning methods [11–14] reduce parameters further without significant performance loss, and small subsets of model weights have been shown to significantly impact model performance [15], collectively highlighting model overparameterization. Our work explores the extreme lower bound of parameter efficiency, demonstrating substantial gains from training as few as 10-100 randomly selected parameters, far fewer than existing approaches.

**Scaling laws and information-theoretic bounds.**   Recent work on scaling laws characterizes performance improvement with increased compute, data, or total model parameters [2, 16, 17].

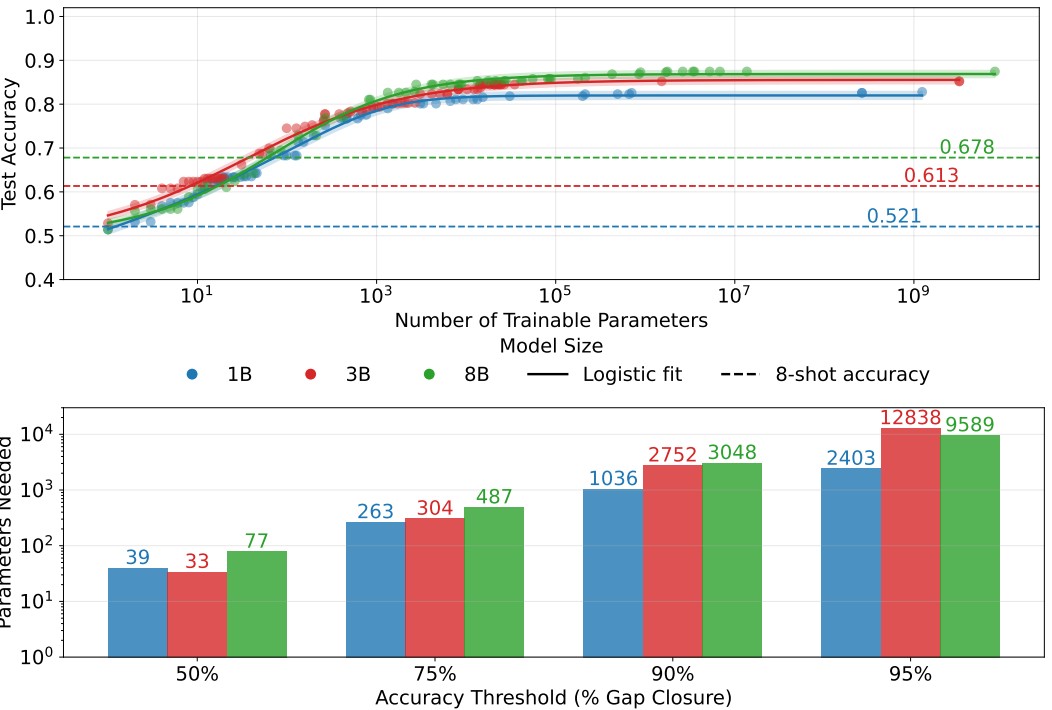

Figure 1: **Fine-tuning only 10-100 randomly selected parameters recovers 50% of performance gap between zero-shot and fully fine-tuned models.** *(Top)* Logistic fits (solid lines) to Pareto frontier points (circles) with bootstrapped 95% confidence intervals (shaded regions) for Llama 3 models of sizes 1B, 3B, and 8B parameters fine-tuned on GSM-8K-CoT-Choice. Dashed lines indicate the 97.5 percentile eight-shot accuracy for each model. *(Bottom)* Minimum number of trained parameters at which various levels of gap closure (50%, 75%, 90%, and 95%) are met or exceeded for each model size (1B, 3B, and 8B parameters).

Complementing this, our "elicitation frontier" explicitly focuses on the minimal parameter budget required for task adaptation. Our parameter-based approach builds on top of Rissanen's Minimum Description Length (MDL) [3] principle and related work [18] that relates the utility of a particular model capability during training to the amount of information contained in the training dataset.

**Lottery tickets and random-parameter hypotheses.** The Lottery Ticket Hypothesis [19] demonstrates that sparse subnets ("winning tickets") can match dense performance but requires expensive search. Our findings show random-parameter selection achieves nearly identical Pareto frontiers without specialized search, suggesting that adaptation information is both compact and distributed throughout the model.

## 3 Methodology

### 3.1 Base models & PEFT setup

In our experiments, we fine-tune a randomly chosen subset of a model's parameters to convergence on different tasks using their public train sets, and we measure how performance on held-out data varies with the minimum number of parameters needed to specify the updated weights of the fine-tuned model.

**Models and baselines.** We fine-tune widely used language models from the Llama and Qwen families (Llama 3.2 1B/3B, Llama 3.1 8B, Qwen2.5-1.5B) [20, 21]. The models used in our experiments are all pretrained base models, unless otherwise noted.

We compare with full fine-tuning, which trains all parameters in the model and defines each model's upper performance ceiling, zero-shot accuracy or loss (performance floor) and eight-shot accuracy (prompt-based elicitation, where applicable) for each model, as well as standard LoRA. In binary classification tasks (BoolQ, GSM-8K-CoT-Choice), we also compare performance with that of a trained, task-specific classifier head on top of the unembedding layer of the base model, as well as a linear probe trained directly on the representations of the middle two layers of the model. Additional details can be found in Appendix A.

**Parameter selection and fine-tuning.** We select random subsets of LoRA parameters (rather than performing structured choice or searching for optimally efficient parameters) as random selection requires no prior information about the model's internal representations. This eliminates the possibility that additional information could effectively be used for elicitation but remain unaccounted for. In experiments that train random subsets of parameters within the LoRA modules, learnable subsets of the LoRA adapter matrices are trained while the remaining adapter weights are held frozen at their initialized values. We select these trainable parameters using uniform random sampling across all LoRA modules in all layers, with each parameter having equal probability of selection regardless of its position in the network. This selection is implemented by applying a random binary mask over gradients during the backward pass, preserving gradient updates only for the selected parameters.

All fine-tuning experiments use AdamW as the optimizer [22], except in experiments training fewer than 20 parameters, where we also perform Bayesian optimization directly on model parameters [23], which often outperforms gradient descent and offers a ~100-500x improvement in training time (GPU hours). However, we use gradient descent for all MDL calculations as prequential coding relies on first epoch cross-entropy loss, which is not computed in Bayesian optimization.

In all experiments, we train until convergence, regardless of the number of epochs required. To verify that only the intended number of learnable parameters has been updated during training, we compare the weights of the LoRA modules at initialization to the weights following fine-tuning and ensure that only the unmasked parameters have been updated. See Appendix C for additional details about fine-tuning procedure details and hyperparameter selection.

## 3.2 Datasets

We fine-tune and evaluate on 4 classification tasks and 4 generative tasks: GSM-8K-CoT-Choice (a CoT correctness classification task, see Section D.1 for dataset details), ARC-Easy, ARC-Challenge [24], and BoolQ [25] for classification tasks, and Alpaca [26], TinyStories [27], Lichess chess puzzles, and s1K-Qwen-1.5B[3] for generation tasks. For reasoning model experiments, we also evaluate on AMC24 and AIME25 (competition math problems), both of which were released after the release of Qwen2.5, to ensure that test set questions cannot have appeared in train data. For all three datasets, we sample 16 responses per question (temperature=0.6) to estimate pass@1 and majority response (maj@16). We also evaluate on AIME24 using the same sampling method to enable comparisons with previous work. For Alpaca, we evaluate the win rate of each fine-tuned model against the best full fine-tuned model when generating answers on Alpaca-Eval (temperature = 0), using Anthropic's internal preference model used for training Claude Sonnet 4 as a judge.

## 4 Empirical Results

### 4.1 Fine-tuning 100 *randomly-chosen* parameters often achieves over 50% performance gap closure on classification tasks

**Elicitation Pareto frontier.** Following the procedure of Appendix E, we select a sequence of trainable parameter budgets spanning from a single parameter to the full set of LoRA adapter weights for ranks 1-1024, in addition to all parameters in the full model (full fine-tuning). We define the empirical frontier as the average accuracy achieved across all seeds for the best set of hyperparameters for each parameter budget. Figure 1 (top) plots the resulting frontier points for GSM-8K-CoT-Choice on a log-scale x–axis, along with the best-fit generalized logistic curve and corresponding bootstrapped 95% confidence intervals.

---

[3]s1K-Qwen-1.5B is an adaptation of s1K [28] which uses prompts from the s1K dataset with completions generated by DeepSeek-R1-Qwen-1.5B (DeepSeek R1 distilled into Qwen2.5-Math-1.5B).

**Gap closure.**    To quantify how many parameters are needed for common performance targets, we compute the budgets at which the given performance gap recovery thresholds are met, with the extent of performance gap defined as the difference between the full fine-tuned model's performance (or maximum of the logistic fit, whichever is higher) and its zero-shot accuracy.

Figure 1 (bottom) reports these values for the three model sizes. Table 1 reports these values for the 50% and 90% thresholds across all multiple-choice datasets and Llama models, comparing them with the performance of zero-shot evaluation on the corresponding pretrained-only base models and that of the full fine-tuned models with all ~$10^9$ parameters trained.

Only ~30 parameters are required to recover 50% of the zero-shot to full-fine-tune gap on GSM-8K-CoT-Choice and BoolQ (Figure 10), meaning that adjusting exclusively a few dozen randomly selected dimensions across all layers of the model can achieve half of the performance gain of full fine-tuning.

The grade-school level science datasets ARC-Easy and ARC-Challenge show a similar overall pattern, albeit with higher parameter requirements than GSM-8K-CoT-Choice, likely due to the more specialized scientific knowledge they assess. In particular, fine-tuning on ARC-Challenge, composed entirely of questions answered incorrectly by both a retrieval-based algorithm and a word co-occurrence algorithm [24], requires significantly more parameters for the smallest model (Llama 3.2 1B) than on other tasks as well as compared to larger models (Llama 3.2 3B and Llama 3.1 8B), potentially indicating capability acquisition rather than elicitation due to the possible absence of the capability in the 1B model following pretraining. We discuss a possible interpretation of this behavior in Section 5.

We observe that larger models also require only few-parameter adjustments to achieve substantial gap closure (Table 1), supporting the hypothesis that larger models already possess most of the requisite knowledge and require minimal tuning to effectively demonstrate it. On all multiple-choice datasets, including both ARC datasets, Llama 3.1 8B achieves 50% performance gap recovery with fewer than 100 trainable parameters.

Table 1: **Number of trainable parameters required to close various performance (accuracy) gap thresholds on multiple-choice tasks** (GSM-8K-CoT-Choice, ARC-Easy, ARC-Challenge). The extent of the full performance gap is defined as the difference between the performance of the fine-tuned model with all parameters trained (Full FT) and the zero-shot accuracy of the pretrained base model. #Params denotes the minimum number of trained parameters at which the model's performance matches or exceeds 50% or 90% of the gap above the zero-shot accuracy. %Params denotes the percentage of trained parameters at each threshold relative to the size of the pretrained base model backbone. Results for natural language generation can be found in Section G.1.

| Dataset | Model | Baselines (%Acc.) | | 50% Gap Closure | | 90% Gap Closure | |
|---|---|---|---|---|---|---|---|
| | | **Zero-shot** | **Full FT** | **#Params** | **%Params** | **#Params** | **%Params** |
| **GSM-8K-CoT-Choice** | Llama-3.2-1B | 47.56 | 87.46 | 39 | $2.67 \times 10^{-6}$ | 1036 | $8.38 \times 10^{-5}$ |
| | Llama-3.2-3B | 48.36 | 85.21 | 33 | $1.21 \times 10^{-6}$ | 2752 | $8.56 \times 10^{-5}$ |
| | Llama-3.1-8B | 51.27 | 87.46 | 77 | $9.58 \times 10^{-7}$ | 2403 | $3.80 \times 10^{-5}$ |
| **ARC-Easy** | Llama-3.2-1B | 24.81 | 76.45 | 216 | $1.74 \times 10^{-5}$ | 4144 | $3.35 \times 10^{-4}$ |
| | Llama-3.2-3B | 74.08 | 87.06 | 774 | $2.41 \times 10^{-5}$ | 159k | $4.95 \times 10^{-3}$ |
| | Llama-3.1-8B | 80.46 | 91.27 | 99 | $1.28 \times 10^{-6}$ | 318k | $3.96 \times 10^{-1}$ |
| **ARC-Challenge** | Llama-3.2-1B | 24.21 | 53.78 | 3023 | $2.44 \times 10^{-4}$ | 154k | $1.24 \times 10^{-2}$ |
| | Llama-3.2-3B | 53.81 | 75.01 | 236 | $7.34 \times 10^{-6}$ | 3077 | $9.58 \times 10^{-5}$ |
| | Llama-3.1-8B | 61.11 | 82.66 | 98 | $1.22 \times 10^{-7}$ | 415 | $5.17 \times 10^{-6}$ |

## 4.2    Fine-tuning 1,000 randomly selected parameters achieves 50% gap closure on some generative tasks

When fine-tuning on the Alpaca instruction-tuning dataset, we observe that training a small fraction of parameters (relative to model size) achieves most of the performance benefits of full fine-tuning for all three Llama models (1B, 3B, and 8B), recovering 50% of the full performance gap in both win

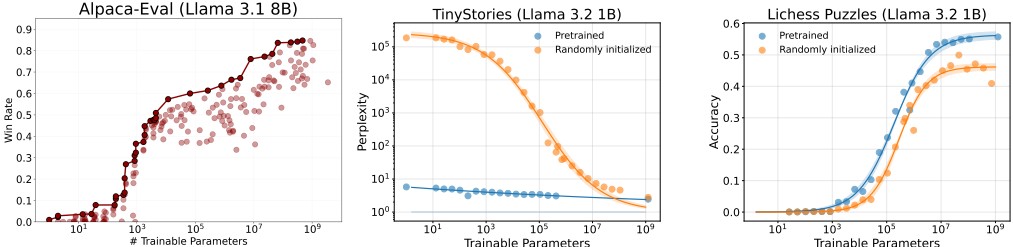

Figure 2: **Fine-tuning small random subsets of parameters can elicit better performance on generative tasks.** Achieving 50% gap closure on Alpaca-Eval (*left*) and TinyStories (*center*) with pretrained Llama models requires training ~1,000 or fewer parameters, whereas ~100,000 parameters or more may required when teaching new, complex capabilities for both story generation (*center*) and optimal chess move prediction (*right*) tasks.

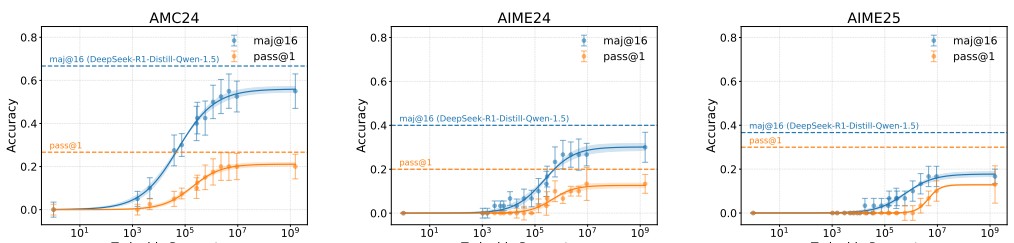

Figure 3: **Distillation of DeepSeek R1-style reasoning scales logistically.** Fine-tuning Qwen2.5-1.5B on 1,000 unique reasoning traces generated by DeepSeek-R1-Distilled-Qwen-1.5B (a post-trained Qwen2.5 model distilled from DeepSeek-R1 using supervised fine-tuning, performance indicated by dashed lines) results in similar trends as observed for Llama models fine-tuned on multiple-choice and generative datasets, suggesting generalization of elicitation scaling behavior to chain-of-thought reasoning tasks and models in other families. For all three datasets, both majority voting (orange) and pass@1 (blue) estimated from sampling 16 responses (temperature=0.6) recover similar logistic scaling with increasing trainable parameters (log-scale x–axis). Error bars indicate standard errors over 16 samples.

rate and loss improvement (Figure 2, left panel, & Figure 7) on Alpaca-Eval with ~1,000 learnable parameters (all models).

When fine-tuning Qwen2.5-1.5B on math reasoning traces from DeepSeek-R1-Distill-Qwen-1.5B, we find it requires a larger number of trainable parameters to close the gap, reflecting the relative difficulty of the task. For both exact match accuracy (maj@16) and loss improvement on AMC24 (over the base model's performance and calculated against reference solutions generated by DeepSeek-R1-Distill-Qwen-1.5B), ~10,000-20,000 trainable parameters are sufficient to achieve 50% gap closure (Figure 3 & Figure 8).

Simpler or easier tasks, such as TinyStories, that make use of capabilities which are already salient in the model largely require fewer trainable parameters to saturate task performance or achieve various performance thresholds, with the necessary parameter budgets appearing correlated with (assumed) task difficulty.

### 4.3   Logistic scaling of elicited performance emerges across tasks and models

We find a logistic relationship between the task performance and the logarithm of tuned parameters, across multiple-choice tasks despite them covering different domains and using different formats (Figure 4), and most generation tasks (Figure 2), including reasoning tasks (Figure 3). While the absolute performance levels and the exact number of random parameters required for specific gap closure thresholds vary across tasks and models sizes, the characteristic S-curve shape in log-parameter space remains consistent.

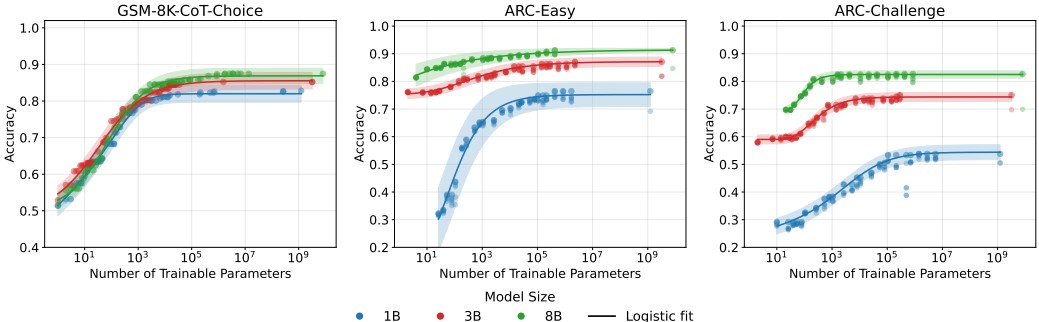

Figure 4: **Logistic scaling patterns emerge across multiple tasks and model sizes.** Each subplot shows model performance (y-axis) for all random seeds on a different task (from left to right: GSM-8K, ARC-Easy, ARC-Challenge) against the number of trainable parameters (x-axis, log scale) for three model sizes (1B, 3B, 8B). Solid lines indicate logistic fits to Pareto frontier points, with shaded regions denoting bootstrapped 95% confidence intervals. Consistent scaling behavior as a function of trainable parameters suggests that general features broadly characterize elicitation patterns and asymptotic model performance across task types, dataset difficulty, and model size.

To verify that the observed logistic relationship is not an artifact of the specific functional form chosen, we evaluate several alternative scaling relationships (power law, saturating exponential, and piecewise linear with power law). Quantitative model comparisons using Bayesian and Akaike information criteria consistently favor logistic fits across all datasets and models (see Appendix F for detailed comparisons). The consistency of these fits across models and tasks, independent of the performance metric used, suggests that the observed scaling behavior reflects an underlying, common relationship between elicited capability level and the logarithm of parameters, rather than being an artifact of our analysis approach.

### 4.4 Stability of Pareto frontier and robustness of logistic scaling across random initializations

Figure 4 shows the distribution of highest-performing models at every parameter budget for every random seed, rather than just the nominal Pareto frontier points. Each point indicates a model fine-tuned with a unique random seed; opaque points correspond to the Pareto frontier, with solid line logistic fits to the frontier-defining points. We observe that the Pareto frontier is insensitive to seed and that the highest performing runs for all seeds overwhelmingly lie within the 95% confidence interval of the fit to the actual frontier.

While there is naturally some variation across random initializations of the LoRA matrices, as well as which modules and layers are adapted, the overall pattern of rapid performance improvement with minimal parameters is consistent. The logistic trend also remains visible across seeds, even with suboptimal hyperparameters (Figure 12), and we observe similar, but shifted, logistic fits independent of optimization effort or method.

### 4.5 Distinguishing elicitation and teaching in pretrained and randomly initialized models

To study qualitative differences between elicitation and teaching, we compare the performance of pretrained and randomly initialized variants of the same model architecture (Llama 3.2 1B) to isolate the effects of a model's relevant, existing task representations on its elicited performance. We focus on two tasks which can be learned without pre-existing language knowledge or capabilities: TinyStories, a simple English short story language modeling task, and mate-in-N chess puzzles sourced from Lichess[4]; both of these tasks use limited vocabularies, require only short context lengths, and have large enough training sets that a 1B parameter model can learn the relevant capability.

For TinyStories, a much easier task for a pretrained model that already understands language, achieving equivalent performance levels (set at 50% gap closure in the pretrained model) requires

---

[4]Mate-in-N chess puzzles reduce the depth of tree search needed to find optimal moves, both simplifying the task of judging a model's win capacity and forcing a more difficult learning curve.

training more than seven orders of magnitude more parameters in the randomly initialized model than in the pretrained model. For Lichess puzzles, a significantly more difficult task that likely requires teaching for both model variants[5], this parameter gap is less than an order of magnitude and 50% gap closure requires more than trainable 100,000 parameters.

# 5 Minimum Description Length

Rissanen's Minimum Description Length (MDL) principle [3] formalizes Occam's razor as the notion that the best model of a dataset is one that provides the simplest explanation, or shortest description, for it. MDL frames algorithmic learning as optimal data compression, in which models that can reconstruct task data from shorter descriptions are said to have learned more about the task.

Perez et al. [18] find that a capability is useful to a model *if and only if* it decreases the length of the model's shortest description of the dataset. By computing a computationally-tractable upper bound on MDL, Rissanen Data Analysis (RDA), it is demonstrated that genuine capabilities always shrink MDL, whereas information accessible to a model that is uncorrelated with the data labels does not.

In our setting, we presume that a capability may be latent in the pretrained (base) model $M_0$ and try to elicit that capability on a dataset designed to surface it. A small adapter $\Delta\theta_k$ with only $k$ trainable weights serves as a pointer to that capability. If a short message (containing the adapter weights $\Delta\theta_k$) lets the model compress the subsequent label stream by many bits, the original base model already likely contained a highly structured representation of the task. Conversely, if the MDL shrinks slowly or not at all until $k$ becomes large, this means training did not find a small pointer to an existing capability, and thus the capability was likely absent and must be learned from scratch rather. MDL compression, given by the difference in description lengths between a frozen pretrained base model and a model with $k$ tunable parameters, $\Delta_k = \mathcal{L}_0 - \mathcal{L}_k$, directly measures how helpful the latent capability is to the model when $k$ parameters are available to specify how to access it, quantifying the intrinsic value of the capability to the task at a specific level of elicitation.

Table 2: An underelicited model versus a model which lacks a capability in MDL terms.

| Elicitation regime | MDL (#Bits to encode labels) | Capability presence |
|---|---|---|
| Training elicits the capability | $\mathcal{L}_k \ll \mathcal{L}_0$ (for $k$ small) | Latent, present after training |
| Training teaches new capabilities | $\mathcal{L}_k \sim \mathcal{L}_0$ (for $k$ small) | Absent or already elicited |

We compute upper bounds on MDL using prequential coding, encoding each example sequentially using the model trained only on previously observed examples, following the procedure of Perez et al. [18]. Practically, we initialize the pretrained model, present each training example exactly once, perform a gradient step immediately after an example (or batch) has been seen, and accumulate the per-token, per-example (per-batch) cross-entropy losses across all examples in the dataset. Full details of this computation and its construction for our particular fine-tuning setting can be found in Appendix I.

## 5.1 Empirical MDL Scaling and Logistic Behavior

For most tasks and models, MDL compression closely tracks improvements in accuracy, smoothly increasing with (randomly selected) trainable parameter count and eventually saturating at a maximal ceiling (Figure 5, top right). Both curves follow logistic forms in log-parameter space.

For Llama 3.1 8B fine-tuned on ARC-Challenge, the derivatives of accuracy and MDL compression nearly coincide and share similar qualitative features, such as near-Gaussian behavior. Figure 5 (top right) demonstrates the close tracking of accuracy with MDL compression (in bits). In contrast, for the same dataset, the accuracy and MDL curves for Llama 3.2 1B, along with their respective derivatives, demonstrate significant discrepancies between both each other and the behavior observed in Llama 3.1 8B (Figure 5, left panels).

To quantify information encoding efficiency, we analyze the derivative of the logistic MDL compression curve versus parameters (Figure 5, bottom). The peak in this derivative indicates the parameter

---

[5]Both remain below 0.01% optimal move accuracy with up to 64 in-context examples.

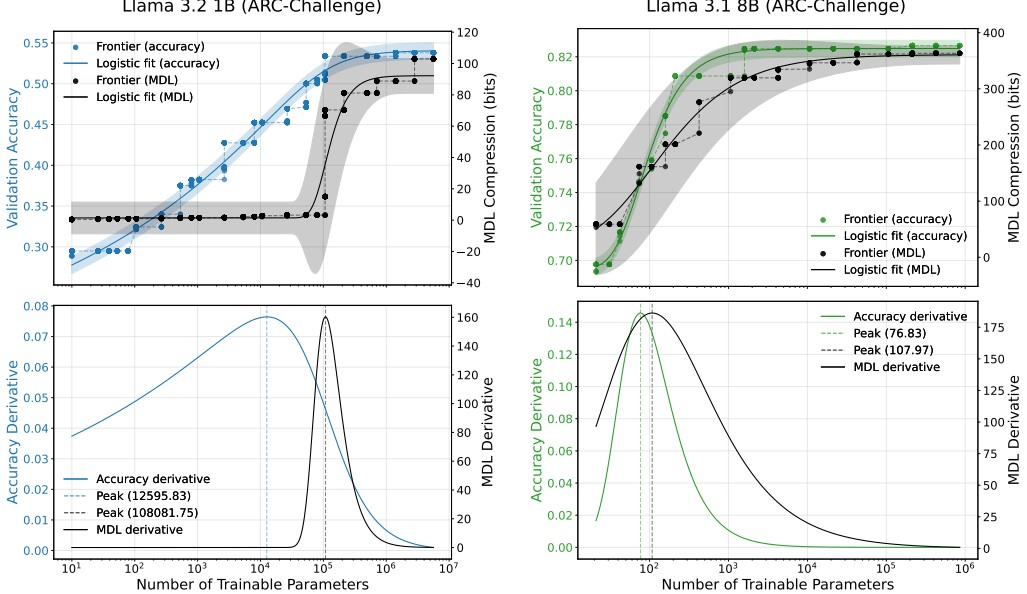

Figure 5: **Accuracy and MDL compression track each other when eliciting latent capabilities but deviate when the required capability is initially absent in the model.** *(Top panels)* Accuracy (colored, left y-axis) and MDL compression (black, right y-axis) plotted against parameter count (x-axis, log scale) for Llama 3.2 1B (left) and Llama 3.1 8B (right) on ARC-Challenge. For 8B, as parameter count increases, MDL compression and accuracy both rise following logistic curves in log-parameter space. In contrast, MDL compression in 1B remains flat up to ~$10^5$ parameters, at which point it discontinuously jumps, though accuracy rises smoothly. Shaded regions indicate bootstrapped 95% confidence intervals on logistic fits. *(Bottom panels)* Derivatives of the accuracy (colored) and MDL compression (black) curves, showing peak rates of change that correspond to highest per-parameter compression and performance efficiency (vertical dashed lines). Both peaks occur at similar parameter counts for Llama 3.1 8B (right) but are separated by approximately an order of magnitude for Llama 3.2 1B (left), indicating significant differences in how efficiently the models are each able to explicitly use the necessary capability to generate simpler models of the dataset, relative to the number of parameters trained.

budget at which marginal additional parameters maximally improve dataset compression. Comparing this to the accuracy derivative reveals subtle differences between information-theoretic encoding efficiency and downstream task performance efficiency: in cases where the pointer to the underlying capability requires many epochs of training to find or generate, MDL compression will be low even if downstream task accuracy is high at the end of multi-epoch training, which may explain why for both Llama 3.2 1B and Llama 3.1 8B the derivative peak is to the right of the parameter peak.

## 6 Discussion

In Section 4 and Section 5, we presented two complementary views of elicitation: an empirical view, where performance rises logistically with the number of trained parameters, and an information-theoretic view, where the Minimum Description Length (MDL) of the label stream falls as soon as a latent capability is unlocked. Here, we synthesise these perspectives and discuss their implications.

### 6.1 Linking performance frontiers and MDL compression

Figure 1 and Table 1 showed that training only ~10-$10^5$ randomly chosen parameters recovers 50–95% of the zero-shot to full-fine-tune gap. Section 5 then established that the same parameter budgets yield hundreds of bits of MDL reduction. Taken together, these results indicate that the logistic frontiers of Section 4 can be viewed analogously to compression curves: each additional parameter

provides the model with an incremental number of bits it can spend on describing the task. The peak of the logistic curve derivative therefore pinpoints an information threshold beyond which adding capacity yields diminishing returns. Models that already contain the requisite capability (e.g. Llama 3.1 8B on ARC-Challenge) reach this threshold quickly, whereas models that cannot readily access a relevant capability (Llama 3.2 1B on the same task), and may need to learn it entirely, require orders of magnitude more parameters before significant reductions in MDL occur.

## 6.2 Practical and safety implications

**Evaluation gaps.** Benchmarks relying on zero-shot or in-context performance may underestimate a model's true capabilities by a large margin. MDL may provide a useful diagnostic: if a small adapter suffices to significantly improve task performance, evaluators should treat the corresponding skill as present, even if accuracy is low in the frozen state or in prompt-based elicitation settings.

**Distributed representations.** The success of random parameter selection suggests that knowledge is spread throughout the network rather than localized in specific layers (Figure 8).

**Logistic-in-log-parameters scaling.** Consistent scaling patterns across tasks and model sizes hints at a possible common property of elicitation that may offer use as a means of predicting future elicitation thresholds for capabilities of interest.

## 6.3 Limitations

- **Task diversity.** While we have evaluated our approach across several classification and generative tasks, none of them have the diversity and complexity of production settings.
- **Theoretical approximations.** Our information-theoretic analysis relies on approximations such as first-epoch loss as a proxy for MDL. More precise formulations that yield tighter upper bounds might reveal additional nuances in the elicitation process.
- **Random selection.** While our random parameter selection approach yields surprisingly strong results, it may not be optimal. Structured approaches to parameter selection or low-probability "winning ticket" parameter selections or initializations could potentially improve efficiency further.
- **Elicitation methods.** There are potentially methods of eliciting capabilities that vastly outperform the current state-of-the-art, making it difficult to provide a tight upper bound on the number of parameters needed for elicitation.
- **Finite dataset size.** The performance we obtain after training $k$ parameters is a lower bound on the maximal performance that is possible to achieve at that capacity, in particular because we only train on datasets of finite (and sometimes relatively small) size. This is especially an issue for large $k$, where the model has more capacity to learn, which might result in our result overestimating gap closure for smaller $k$.

## 7 Conclusion

We present a novel framework for quantifying the elicitation of latent capabilities in language models based on the minimum number of trainable parameters required to achieve various performance levels on downstream tasks. We propose quantifying elicitation with an "elicitation frontier," a curve that maps the number of trained parameters to performance improvements. Across 8 tasks and 4 model sizes from two popular model families, we find that 20-1000 randomly selected parameters suffice to recover most of the performance gap between zero-shot and full fine-tuning.

Our elicitation frontier framework offers a complementary perspective to traditional scaling laws, focusing on the minimal information required to unlock capabilities rather than the maximal capacity of models. The consistent logistic scaling relationship we observe across tasks and model sizes suggests that fundamental, shared principles may govern the ways in which capabilities scale with parameter adjustments. Our study complements classic compute- and data-centric scaling laws with an information-centric perspective: how much must be said to a model before it shows what it already knows? Understanding this information budget is essential for capability forecasting, responsible deployment, and the design of safe, reliable, and predictable AI systems.

# 8 Acknowledgments

J.L. and J.S. proposed the initial idea for the project. E.D. designed and carried out the majority of experiments across all tasks and model sizes and performed the minimum description length (MDL) calculations. H.J. conducted the Alpaca experiments and contributed all related analyses. J.L., F.R., and E.P. provided supervision at various stages. E.D. wrote the paper with input from F.R., H.J., and J.L. E.P., M.R.D., J.M., J.S., and A.S. offered feedback on early versions of the manuscript. H.S. provided logistical and organizational support. We gratefully acknowledge compute resources provided by Scale and Anthropic. We thank Eric Easley, Vivek Hebbar, Alex Cloud, Joe Carlsmith, Sara Price, Aryan Bhatt, John Hughes, Daniel Paleka, and Joshua Clymer for helpful discussions. M.R.D. acknowledges support from the U.S. Army Research Laboratory and the U.S. Army Research Office under Contract No. W911NF-20-1-0151.

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

# A Baselines

For all tasks, we use three baselines for comparison (zero-shot performance, eight-shot performance, and fine-tuning all parameters of the base model). We define each model's performance floor as its zero-shot accuracy (multiple-choice datasets) or zero-shot win rate (Alpaca). We take each model's highest performance (accuracy or win rate) following full fine-tuning (updating all model weights) as its nominal ceiling performance.

**Full fine-tuning.** For all models, we fine-tune the entire model until convergence on each dataset. We define convergence as the point where validation accuracy exhibits no improvement for 20 consecutive epochs. Specifically, if $|\text{Acc}(t) - \max(\text{Acc}(t - 20 : t))| < \varepsilon$ where $\varepsilon = 10^{-4}$, training terminates. This criterion ensures stable convergence while avoiding premature stopping.

**Zero-shot evaluation.** For each model, we perform zero-shot evaluation on the validation and test sets. Prompts used for zero-shot evaluation are the same as those used for all fine-tuning experiments. Prompt formats for each dataset can be found in Appendix D.

**Multi-shot evaluation.** We perform multi-shot evaluation similarly, using the same prompt format as used for zero-shot evaluation and prepending randomly selected examples (with properly formatted correct responses for each) from the corresponding train set. The number of shots is equal to the number of prepended train set examples with answers. We use the same formatting for each example, prepending the same train set examples each time, with the final example being the validation or test set example that we evaluate the model's response on, in which the answer field is left blank.

We tested variations with 1, 2, 4, 8, 16, and 32 shots prepended for each model and dataset, finding that 8 shots resulted in the highest performance across the board. We average performance across $n = 30$ trials for each number of shots $X$ to obtain the overall $X$-shot performance. Every trial uses a different random seed that is used only to select the specific train set examples that are prepended; the same set of random seeds is used for all models, datasets, and $X$-shot configurations. We ensure that the prepended examples contain an approximately even split of each answer class to reduce potential sources of bias in the model outputs. Due to variation in performance across seeds, we use the 97.5 percentile ($2\sigma$ above the mean) value on the distribution of scores as an approximate upper bound baseline for the model's $X$-shot performance.

## A.1 Additional binary classification baselines

For binary classification datasets GSM-8K-CoT-Choice and BoolQ, we compare the performance of two additional baselines: (i) training a task-specific classifier head attached to the unembedding layer of the model, and (ii) training a linear probe on the representations of the middle two layers of each model. Figure 6 compares these baselines for Llama-3.2-1B on GSM-8K-CoT-Choice.

**Classifier Head.** We initialize each model with a classifier head that enforces the model's outputs to be one of the two label classes. The classifier head contains twice the number of parameters of the model's hidden dimension. We then freeze the model backbone and train only the weights in the classifier head when fine-tuning on each dataset, taking the highest final evaluation performance at convergence across seeds ($s = 5$) as the baseline.

**Linear Probe.** We initialize each model with all parameters frozen and train a linear probe out the representations of the middle two layers of each model.

# B Generative Experiments

## B.1 Alpaca

**Discussion.** Across the Alpaca instruction-tuning runs, win-rate versus trainable-parameter count similarly increases quickly with few parameters (Figure 7). For Llama-3.2-1B, win rate (versus the best full fine-tuned model) jumps from ~0.08 to ~0.71 as the budget grows from $10^3$ to $10^5$ updated weights. The 3B model follows a similar pattern, shifted one decade to the left. Note that overall, the win rate is lower for 8B (compared to 1B or 3B) as it is a higher-capacity base model with relatively higher full fine-tuned performance.

**Dataset and splits.** For the open-ended generation experiments we adopt the Stanford Alpaca instruction-following corpus. The raw 52k examples are shuffled once with a fixed random seed and partitioned 90/10 into training (234,006 instructions) and held-out evaluation (26,006 instructions). No additional filtering or augmentation is applied. All results in the main text use the full training split; the "tiny" ablations described below operate on the same data but restrict the number of trainable parameters, not the dataset size.

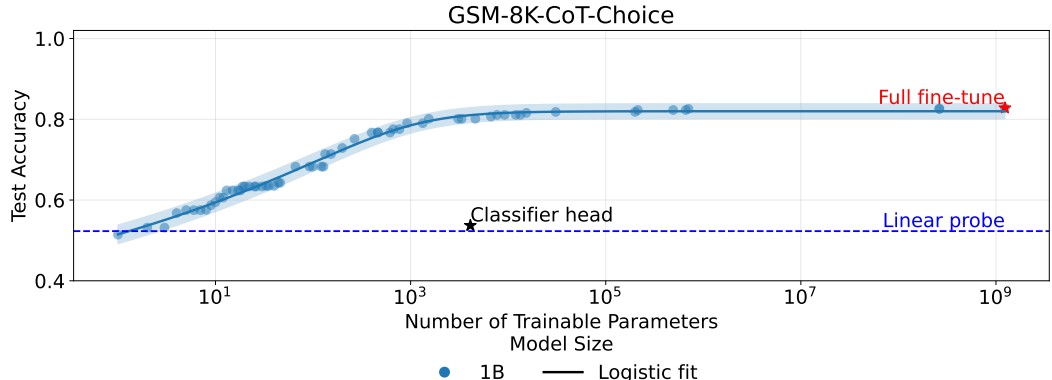

Figure 6: Llama-3.2-1B fine-tuned on GSM-8K-CoT-Choice with full fine-tune (red star), classifier head (black star), and linear probe (blue dashed line) baselines.

**Prompt construction and tokenization.**  Each instance is formatted with the standard Alpaca template where the input block is omitted when empty. The expected answer is appended after the prompt. The entire sequence is truncated to 256 tokens. During training we follow common instruction-tuning practice and compute loss only on the response: all prompt tokens are replaced with the ignore-index so gradients flow exclusively through the generated portion. The end-of-sequence token is reused as the pad token.

**Base models and adaptation strategy.**  We fine-tune two pretrained Llama-3.2 models—1B and 3B parameters—using low-rank adaptation. Unless otherwise noted, LoRA modules (ranks 1, 2, 4, 8, 16, 256, 512) are inserted on every projection matrix in every transformer block.

To study how few parameters suffice for high-quality generation we treat the weights inside the LoRA matrices as an unordered pool and select an absolute number of them to update (1 to full rank). Selection is uniform at random over all LoRA weights; the mask is sampled once at the start of each run and held fixed for the duration of training. In the very-few parameter regime (<10k trainable parameters) the original language-model head remains frozen to isolate the effect of the highly sparse adaptation; at larger budgets the head is fine-tuned together with the adapters.

**Optimization and runtime.**  Every model is trained for exactly two passes over the 234k-example training split. Mixed-precision (bfloat16) training with FlashAttention-2 is employed throughout, and gradient check-pointing is enabled except in the smallest runs. Each experiment fits on a single NVIDIA H100 GPU; no multi-GPU or distributed training was employed.

**Evaluation protocol.**  After fine-tuning we generate completions for the 805-example Alpaca evaluation subset using greedy decoding (temperature 0, max 256 tokens). Outputs are scored by an off-the-shelf preference model that compares each fine-tuned model's answer against the corresponding answer from the un-adapted base model of the same size. We report the win rate of the fine-tuned model over the 805 comparisons; ties and losses are counted separately but not shown in the main tables.

**Summary of reported configurations.**  Model sizes: Llama 3.2 1B, Llama 3.2 3B, Llama 3.1 8B

LoRA ranks: 1, 2, 4, 8, 16, 256, 512

Trainable-parameter budgets: exact counts spanning from just a few parameters to nearly full finetuning.

Epochs: 2 (all runs)

Sequence length: 256 tokens

Hardware: single NVIDIA H100 80 GB GPU per run

## B.2 Reasoning distillation with s1K rollouts

To determine the parameter efficiency of eliciting reasoning capability in Qwen base models, we generate reasoning traces for all prompts in the s1K dataset [28] using the respective post-trained (DeepSeek-R1 distilled) version of each model. The post-trained models used were originally trained by DeepSeek using supervised fine-tuning (SFT) on a large corpus of several hundred thousand reasoning traces generated by DeepSeek-R1. We then perform SFT on each original base model using the dataset of the corresponding post-trained model's

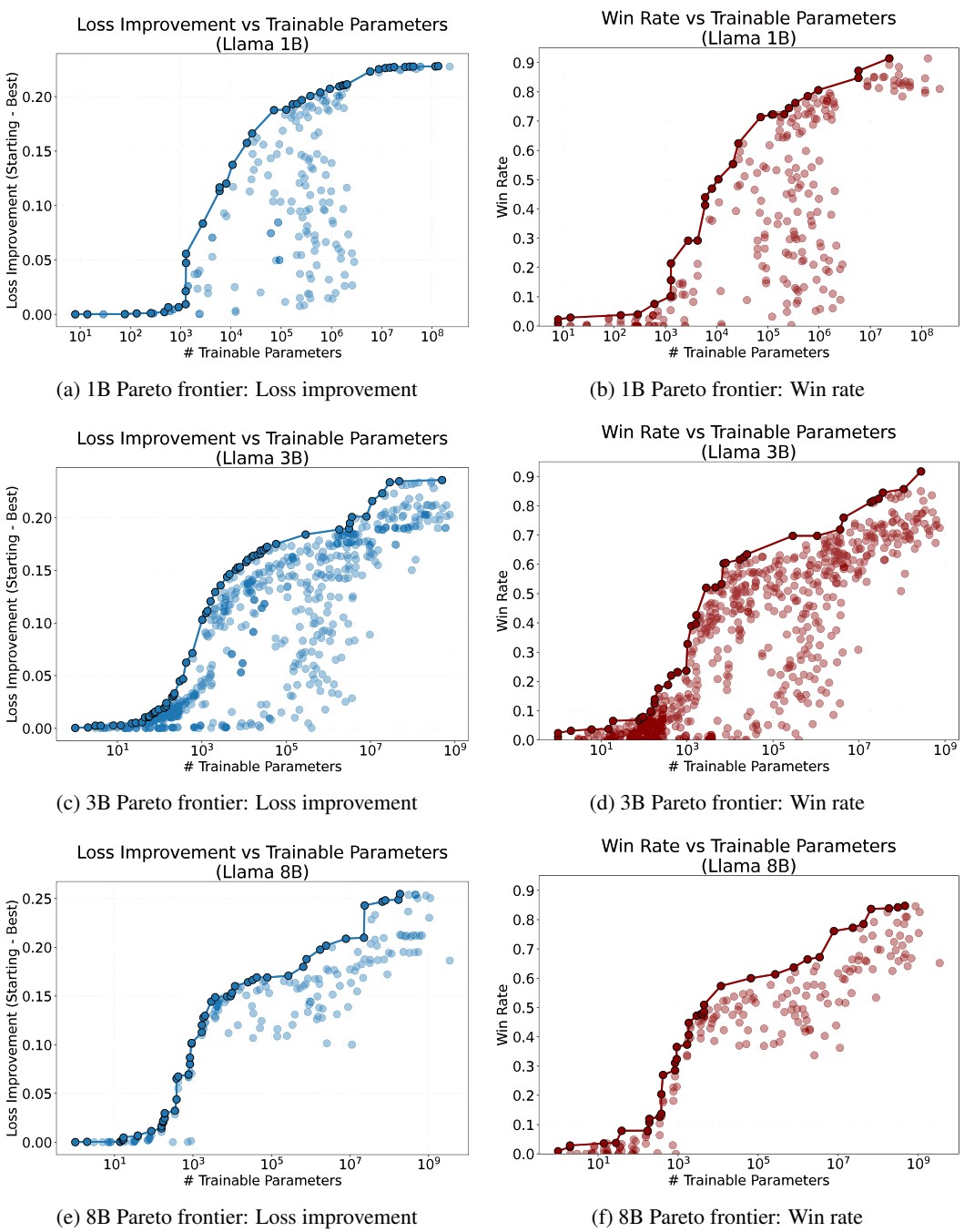

(a) 1B Pareto frontier: Loss improvement

(b) 1B Pareto frontier: Win rate

(c) 3B Pareto frontier: Loss improvement

(d) 3B Pareto frontier: Win rate

(e) 8B Pareto frontier: Loss improvement

(f) 8B Pareto frontier: Win rate

Figure 7: Comparison of loss improvement and win rate for Llama models fine-tuned on Alpaca and evaluated on AlpacaEval (1B, 3B, and 8B). Loss improvement is computed as each fine-tuned model's (blue points) improvement in the cross-entropy loss for autoregressive generation compared to its corresponding base model. Win rate denotes the proportion of responses generated a fine-tuned model (red points) that are preferred by the reward model over responses generated by the best full fine-tuned model. In each graph, single points denote the result of a single fine-tuned model with the specified number of trainable parameters. The Pareto frontier in each graph is denoted by the opaque points with black outline connected by a solid line.

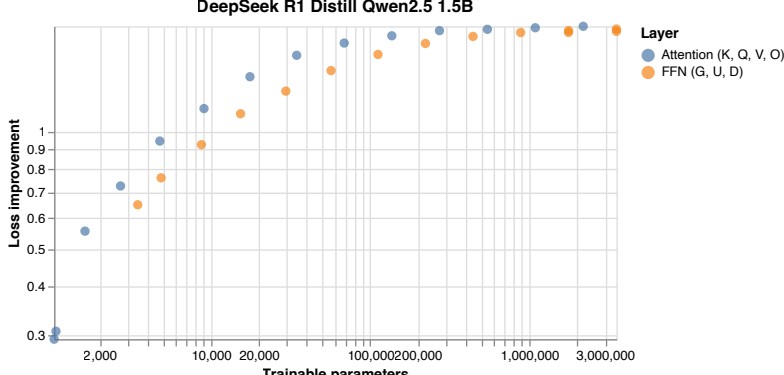

Figure 8: Adapting $K, Q, V, O$ matrices in the self-attention layers (blue points) is more parameter-efficient for (sparse random-parameter) LoRA adaptation than adapting $G, U, D$ matrices in the feed-forward network/MLP layers of the transformer block (orange points). For the same trainable parameter budget, adapting weight matrices in the self-attention layers results in higher loss improvement of the converged, fine-tuned model over the base model's zero-shot loss (results shown for evaluation on the AMC23 dataset). This effect is particularly pronounced when fine-tuning sparse random-parameter subsets of the rank-1 LoRA adapters. Differences in loss improvement are similar when training all adapter parameters (i.e., "full-rank"), nearly independent of rank and regardless of whether the self-attention or FFN layers in each transformer block are adapted.

generations to the s1K prompts; we use the same method for (sparse) LoRA adaptation and full fine-tuning as described in Appendix C.

## C Fine-tuning details

### C.1 Target Modules

We investigate variations in which we adapt all weight matrices within the transformer block, $Q, V$ or $Q, V, K, O$ matrices in the self-attention layers, $G, U, D$ matrices in the feed-forward network (FFN)/MLP layers, $K, Q, V, U, D$ matrices across the transformer block, and randomly selected subsets of weight matrices across the transformer block. We observe similar logistic scaling behavior of performance versus number of learnable parameters across all configurations of adapted weight matrices.

We find that attention-only tuning (adapting $K, Q, V, O$ matrices) is consistently the most parameter-efficient across all datasets (multiple choice and generative, including reasoning), irrespective of answer format. This suggests that the information needed to activate pre-existing knowledge is not confined to the FFN subspace. Figure 8 compares the loss improvement for Qwen-2.5-Math-1.5B fine-tuned on 1,000 reasoning traces generated from R1-Distill-Qwen-1.5B[6], a model distilled from DeepSeek-R1 into the same Qwen-2.5-Math-1.5B base model via supervised fine-tuning.

When we refer to "trainable parameters," we exclusively mean scalar values within LoRA adapter matrices (except in full fine-tuning, in which we directly train the base model's original parameters). For rank-$r$ adaptation:

- Matrix A: $\mathbb{R}^{r \times d_{in}}$ (Kaiming uniform initialization)
- Matrix B: $\mathbb{R}^{d_{out} \times r}$ (zero initialization)
- Total parameters per module: $r \times (d_{in} + d_{out})$

### C.2 Seed and hyperparameter variation

For each parameter budget, we run up to 5 random seeds with up to 50 Bayesian-Hyperband trials each over learning rate, batch size, and weight decay. In all experiments, we train until convergence (defined as at least 20 epochs with no improvement in validation accuracy for multiple choice tasks, or at least 2 epochs with no improvement in validation loss for generative tasks), regardless of the number of epochs required.

---

[6]Prompts used for generating the model reasoning trace rollouts used for supervised fine-tuning were derived from the s1K dataset [28].

**Search spaces.** Parameter search ranges are calibrated based on the number of trainable parameters and GPU memory constraints:

- Learning rate: $[10^{-6}, 1]$ (log-uniform spacing)
- Batch size: $\{1, 2, 4, 8, 16, 32, 64, 128, 256\}$ (discrete)
- Weight decay: $[0, 0.1]$ (uniform sampling)

## C.3 Multiple-choice scoring

For every multiple-choice dataset, we score the model by directly comparing logits of the canonical answer tokens (e.g., "A"/"B"/"C"/"D") at the final position.

To reduce the dependence of the results on the model's ability to format its responses properly, we additionally run an ablation that sums probabilities over several token variations per option (including different capitalization, synonyms, whitespace). We then calculate loss and accuracy directly using the logits for tokens corresponding to each label class.

For multiple-choice tasks, we assign labels by aggregating probabilities across variations for each ground-truth label token:

$$P(\text{label}) = \sum_i P(\text{token}_i), \tag{1}$$

where $\text{token}_i \in \{\text{"A", " A", "a", " a"}\}$ for label A (and similarly for all other labels).

We take the softmax of the model's output vector for the final token position to obtain probabilities on the outputs for each token in the model's vocabulary, and for each label class, we sum the probabilities of differently formatted, single-token variations of each label (e.g., "A": ("A", " A", "a", " a") for label class A). We use the same set of variations for all label classes (e.g., "Correct": ("Correct", " Correct", "correct", " correct"), "Incorrect": ("Incorrect", " Incorrect", "incorrect", " incorrect") with sets of labels for each class incorporating the same variations on spacing and capitalization). This approach ensures robustness to minor formatting variations in the model's output while maintaining evaluation consistency.

The two procedures (logit comparison on ground-truth labels only versus cumulative probability comparison on variations of ground-truth labels) differ by <0.2 percentage points on average, demonstrating that memorization of the output format does not explain the observed increases in performance.

## C.4 Bayesian optimization directly over LoRA adapter weights (ultra-low parameter regimes)

For experiments in which fewer than 20 parameters are trained, training until convergence can require more than 1000 training epochs, and training dynamics can become unpredictable with discontinuities observed in the loss. The combination of these factors makes it difficult to identify when training has converged or to explore sufficient hyperparameter configurations to effectively resolve the true elicitation Pareto frontier.

To circumvent this issue, in addition to training with gradient descent, in the regime of fewer than ~20-25 trainable parameters, we also perform Bayesian optimization with a Gaussian process surrogate model and expected improvement acquisition function (GPyTorch with default Matérn kernel) directly on the learnable adapter weights within the LoRA matrices; parameter selection is performed using the same random sampling procedure as in the gradient-descent-based fine-tuning setting. This approach mitigates optimization instabilities at extreme sparsity levels.

For the Gaussian process, parameter bounds were empirically calibrated through preliminary gradient descent experiments:

- $k \leq 2 : [-10, 10], [-5, 5]$
- $3 \leq k \leq 10 : [-5, 5], [-3, 3]$
- $11 \leq k \leq 25 : [-3, 3], [-1, 1],$

where $k$ is the number of trainable parameters. The Gaussian process optimizer directly searches this k-dimensional space, evaluating model performance for each parameter configuration. The same bounds are used for all trainable weights (uniform bounding hypercube). We initially uniformly randomly sample from within the bounding hypercube a set of candidate vectors that contains three times as many vectors as the number of trainable weights. Each vector has dimension equal to the number of trainable weights, with each entry in a vector corresponding to an individual trainable weight.

For each candidate vector, we replace the values of the trainable weights in the original LoRA matrices with the corresponding candidate entries and evaluate on the entire train and validation sets. We use the initial set of candidate vectors with evaluated accuracies to seed the Gaussian process optimization. We iterate over all candidate vectors, generating up to 400 additional, new candidate vectors with the Gaussian process optimizer, and take the performance from the candidate vector with highest validation accuracy as the final performance (with weights of the final trained parameters replaced with that candidate vector's entries).

This process entirely eliminates the need for hyperparameter search and significantly reduces training time and overhead, as only the forward pass must be computed for optimization. In this very-few parameter limit, we find that performance from models trained with Bayesian optimization matches or exceeds the performance of the best performing hyperparameter configurations, enabling us to efficiently sample the elicitation frontier.

Due to the fact that Bayesian optimization does not employ training epochs, however, we are unable to calculate prequential minimum description lengths for the resulting fine-tuned models. Instead, we calculate prequential MDL only for models fine-tuned using gradient descent.

## C.5    Computational Requirements

- Hardware: Single NVIDIA H100 80 GB GPU
- Runtime: 10 minutes to 36 hours depending on parameters trained and total training epochs
- Memory: 20-60 GB depending on model size and batch size
- Total experiments: ~3500 individual runs

# D    Dataset details and fine-tuning prompts

## D.1    GSM-8K-CoT-Choice

GSM-8K-CoT-Choice was created by adapting ~1000 questions from the GSM-8K benchmark [29] to a binary classification task. For each question, four distinct chain-of-thought reasoning solution attempts and answers were generated with Claude 3.5 Sonnet. In each example, the model is presented with a question, a solution attempt, and a final answer and asked to identify whether the solution attempt and final answer are correct. The dataset is split evenly between correct and incorrect solutions, with two of each per question. During development, one example from each label class per question (one true, one false) was randomly sampled to construct the fine-tuning dataset, and we confirmed similar training dynamics across random seeds. Model-generated solution attempts were rigorously validated by a human and additional language model (Claude 3.5 Haiku) to ensure accurate labeling.

### D.1.1    Dataset generation procedure for GSM-8K-CoT-Choice

We use the following procedure to generate the dataset:

1. **Question selection:** Sample 1,000 problems from GSM-8K training and validation sets
2. **Solution generation:** For each problem, generate 4 chain-of-thought solutions using Claude-3.5-Sonnet with structured JSON output (2 correct examples, 2 incorrect examples)
3. **Answer validation:**
   - Correct solutions: Verify exact string match with ground-truth answer
   - Incorrect solutions: Ensure no match with ground-truth answer
4. **Quality control:** Manual verification of flagged examples where automated validation fails
5. **Balance Verification:** Confirm exactly 2 correct and 2 incorrect solutions per problem
6. **Sampling:** Randomly sample one example from each label class per question

### D.1.2    Dataset statistics for GSM-8K-CoT-Choice

- Total examples: 4,000 (1,000 questions $\times$ 4 solutions each)
- Class balance: 50% correct, 50% incorrect
- Token length: 99% of examples < 384 tokens (using Llama tokenizer)

## D.2    DeepSeek-R1 distilled model generations to s1K prompts

The dataset used for supervised fine-tuning each base model was created by generating rollouts (model generations) to each prompt in the s1K dataset [28] using the official DeepSeek-distilled version of the specific base model, up to a maximum length of 32768 tokens (temperature=0.6). Following the procedure of Muennighoff et al. [28], we verify that the reasoning traces have the correct format (i.e., a thinking portion where initial reasoning is carried out, followed by an answer to the user query) but do not regenerate incorrect solutions.

## D.3    Fine-tuning Prompts

Systematic evaluation revealed minimal sensitivity to prompt variations when instructions are present (<0.5% performance difference). Without instructions, performance degrades by approximately 2%, motivating our standardized prompt templates.

### D.3.1   GSM-8K-CoT-Choice

```
You will be given a math problem, a step-by-step solution attempt,
and a final answer.
Evaluate the correctness of the solution attempt and answer to the
problem.
Respond with ONLY 'Correct' or 'Incorrect.'
Problem: {problem}
Solution Attempt: {solution attempt}
Answer: {answer}
Respond with ONLY 'Correct' or 'Incorrect'. Response:
```

### D.3.2   ARC-Easy & ARC-Challenge

```
You will be given a question and a series of answer choices labeled
A, B, C, D, etc.
Select the correct answer from the choices.
Respond with ONLY the uppercase letter of the correct answer (A-E).
Question: {question}
Answer choices:
A: {choice A}
B: {choice B}
... {other choices}
Response:
```

### D.3.3   BoolQ

```
You will be given a passage and a question.
Determine the answer to the question based on the information in the
passage.
Respond with ONLY 'Yes' or 'No.'
Passage: {passage}
Question: {question}
Response:
```

### D.3.4   Alpaca

```
Below is an instruction that describes a task{, paired with an input
that provides further context}.
Write a response that appropriately completes the request.

### Instruction:
{instruction}

### Input:
{input}

### Response:
```

# E   Preliminaries

**Notation.**   Let $\mathcal{D} = \{(x_i, y_i)\}_{i=1}^{n}$ denote a downstream task dataset of size $n$. We denote by $\theta_0$ the parameters of the pretrained base model and by $\theta_{p,s,h}$ the LoRA-adapted parameters ontained by training exactly $p$ adapter weights with random seed $s$ and hyperparameters $h$. We write

$$\text{Acc}(\theta) \;=\; \frac{1}{|\mathcal{D}_{\text{val}}|} \sum_{x,y \in \mathcal{D}_{\text{val}}} \mathbf{1}\{\hat{y}_\theta(x) = y\} \tag{2}$$

for the validation accuracy of the model $\theta$. We further define the zero-shot baseline $\text{Acc}_0 = \text{Acc}(\theta_0)$ and the accuracy of the fully fine-tuned model $\text{Acc}_{\text{full}}$.

**Parameter Budget Selection.**    We select parameter budgets $\{p_1, ..., p_k\}$ on a logarithmic scale to ensure adequate resolution across orders of magnitude.

**Pareto Frontier Extraction.**    To characterize the trade-off between parameter budget and performance, we choose a set of budgets $p \in \{p_1, \ldots, p_K\}$. For each budget $p$, we perform:

- $S = 3-5$ independent random seeds $s$.
- For each $(p, s)$, up to 50 hyperparameter trials $h$ via Bayesian optimization with Hyperband [30].
- Record the maximal validation accuracy

$$f(p) \;=\; \max_{s,h} \; \text{Acc}\big(\theta_{p,s,h}\big). \tag{3}$$

The set of points $\big\{(p,\, f(p))\big\}_{p=1}^{K}$ defines the empirical Pareto frontier. We fit a generalized logistic in $\log p$ for its flexibility in modeling diverse S-shaped growth patterns:

$$f(p) \;\approx\; \text{Acc}_\infty \;-\; \frac{\text{Acc}_\infty - \text{Acc}_0}{\left(1 + \exp\left(-a - b \log p\right)\right)^\nu}, \tag{4}$$

using nonlinear least squares to recover $(\text{Acc}_0, \text{Acc}_\infty, a, b, \nu)$. This formulation accommodates asymmetric growth curves observed across different tasks.

**Bootstrap Confidence Intervals.**    We employ nonparametric bootstrap with $10^4$-$10^5$ samples:

1. For each iteration, remove one frontier point uniformly at random
2. Fit logistic curve to remaining points
3. Compute 95% confidence intervals from empirical distribution of fitted parameters

**Empirical Metrics.**    We evaluate two additional quantities on the frontier models $\theta_{p,s^*,h^*}$ achieving $f(p)$:

1. *Gap closure*:

$$\text{GC}(p) = \frac{f(p) - \text{Acc}_0}{\text{Acc}_{\text{full}} - \text{Acc}_0} \;\in [0, 1].$$

2. *Online MDL* [3, 31]:

$$\text{MDL}(p) = \sum_{i=1}^{n} \ell\big(\theta_{p,s^*,h^*};\, x_i, y_i\big),$$

where $\ell$ is the per-example training loss, which is updated after each example (or batch) has been processed, and the sum is taken over the first epoch only, such that each train set example has been seen exactly once by the model.

# F   Analysis of alternative fitting functions

## F.1   Alternative fitting functions

We analyze four alternative fitting functions in addition to a generalized logistic: saturating exponential, power law, (piecewise) broken power law, and smooth broken power law (power law multiplied with a logistic window to maintain differentiability).

Generalized logistic:

$$y = y_{\min} + \frac{y_{\max} - y_{\min}}{(1 + e^{-k(x-x_0)})^\nu} \tag{5}$$

Saturating exponential:

$$y = y_{\max} - Ae^{-B(x-x_0)} \tag{6}$$

Power law:

$$y = ax^b \tag{7}$$

Broken power law:

$$y = \begin{cases} a + bx & \text{for } x \leq x_k \\ a + bx_k + c(x - x_k)^d & \text{for } x > x_k \end{cases} \tag{8}$$

Smooth broken power law:

$$y = \begin{cases} a + bx & \text{for } x << x_k \\ a + bx_k + c(x - x_k)^d & \text{for } x >> x_k \end{cases} \tag{9}$$

### F.2 Model comparison metrics

We evaluate model fit using:

- Akaike information criterion (AIC): $2k - 2\ln\hat{L}$
- Bayesian information criterion (BIC): $k\ln n - 2\ln\hat{L}$
- Mean square error (MSE): $\frac{1}{n}\sum_i(y_i - \hat{y}_i)^2$,

where $k$ is the number of fitting parameters (not trainable parameters in this instance), $n$ is the number of data points (i.e., the number of Pareto frontier points used for generating the fitted curve), and $\hat{L}$ is the maximized likelihood (sum of squared errors). Figure 9 shows the results of curve fitting for all three Llama model sizes (1B, 3B, and 8B) fine-tuned on GSM-8K-CoT-Choice with corresponding AIC and BIC for each fit.

## G Results from additional datasets

### G.1 Alpaca

Figure 7 shows results from fine-tuning Llama-3.2-1B and Llama-3.2-3B on the Alpaca instruction-tuning dataset.

### G.2 BoolQ

Figure 7 shows results from fine-tuning Llama-3.2-1B and Llama-3.2-3B on BoolQ. The logistic scaling of accuracy with increasing trainable parameters remains consistent in BoolQ, with both models requiring ~100 parameters to achieve 50% performance gap closure between zero-shot accuracy and full fine-tune accuracy.

### G.3 Dataset Coverage Limitations

BoolQ and Alpaca experiments include only Llama-3.2-1B and 3B models due to computational constraints. The consistent logistic scaling observed across completed experiments suggests similar patterns would emerge for the 8B model, though empirical validation remains future work.

## H Results across hyperparameter and seed variations

Because we fine-tune with numbers of trainable parameters spanning several orders of magnitude, we optimize hyperparameters at each parameter budget for each seed, performing up to 50 trials of Bayesian optimization with Hyperband. Figure 12 shows results for hyperparameter optimization trials for Llama 3.2 1B (top) and Llama 3.2 3B (bottom) fine-tuned on ARC-Easy (all seeds). Hyperparameter optimization is required for each parameter budget as hyperparameters (learning rate, in particular) can differ significantly depending on the sparsity of trainable parameters. For the highest performing seeds, as shown in Figure 4, the highest performing runs for each seed demonstrate little variability in final validation accuracy and largely fall within the 95% confidence interval of the fit to the Pareto frontier.

## I Construction of Prequential MDL and RDA

Because the true Minimum Description Length is uncomputable, Perez et al. [18] propose to upper bound it with the data's MDL obtained by prequential coding:

$$\mathcal{L}(y_{1:N} \mid x_{1:N}, f) < \mathcal{L}(y_{1:N} \mid x_{1:N}), \tag{10}$$

which states that a model that uses a domain-relevant capability to generate a label obtains a more compact explanation of the label-generation process, which enables it to succinctly model the dataset itself.

We can alternatively view this label-generation learning process as a communication protocol for sending the labels for the dataset from someone who has them to someone who does not. Alice, who has the labels, wants to share them with Bob, who does not, and they want to find the smallest file Alice needs to send to Bob so he can have them. Alice and Bob share the same base model $M$ and learning algorithm $A$ (which contains all

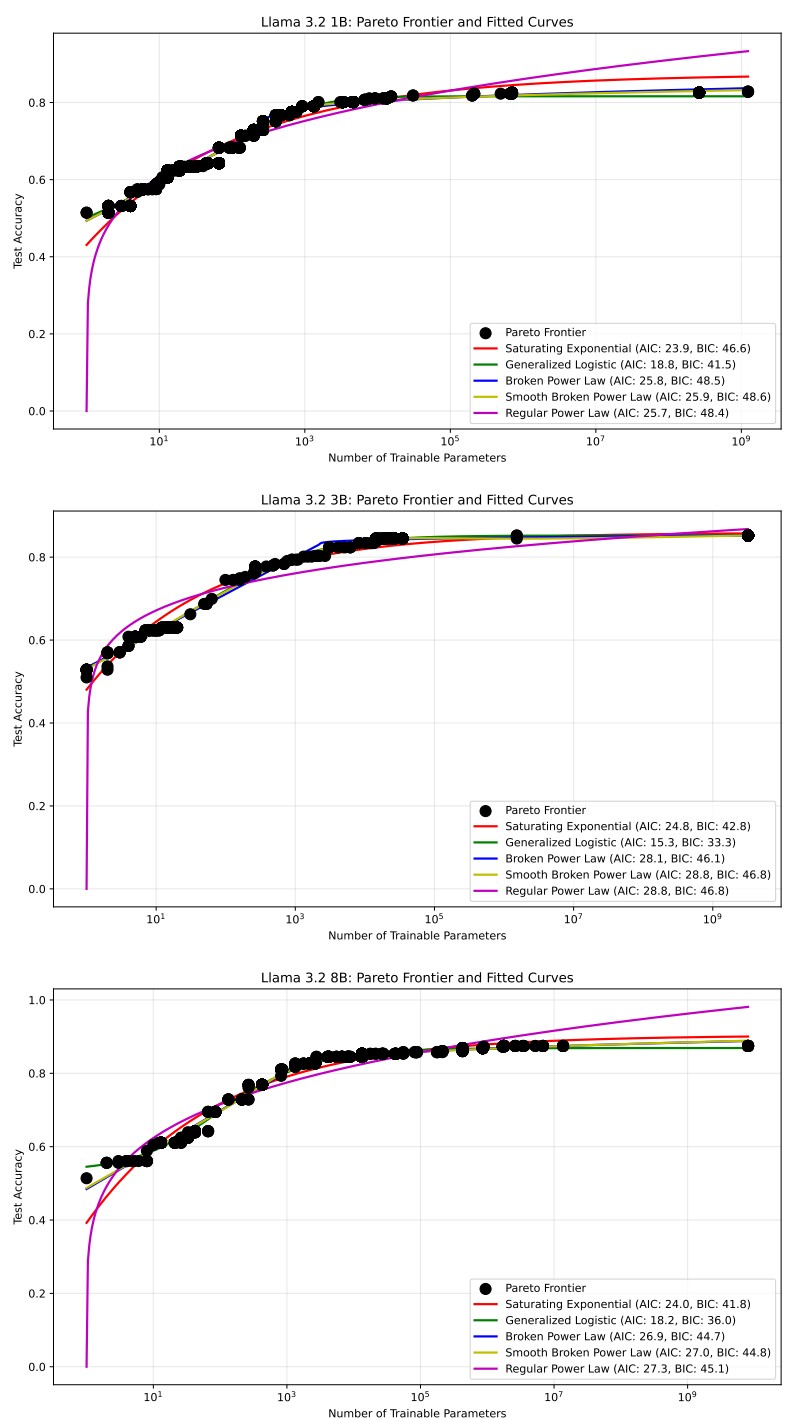

Figure 9: Fitting of curves with corresponding AIC and BIC for 1B, 3B, and 8B Llama models fine-tuned on GSM-8K-CoT-Choice. For all models, AIC and BIC are consistently lowest for fits to a generalized logistic function.

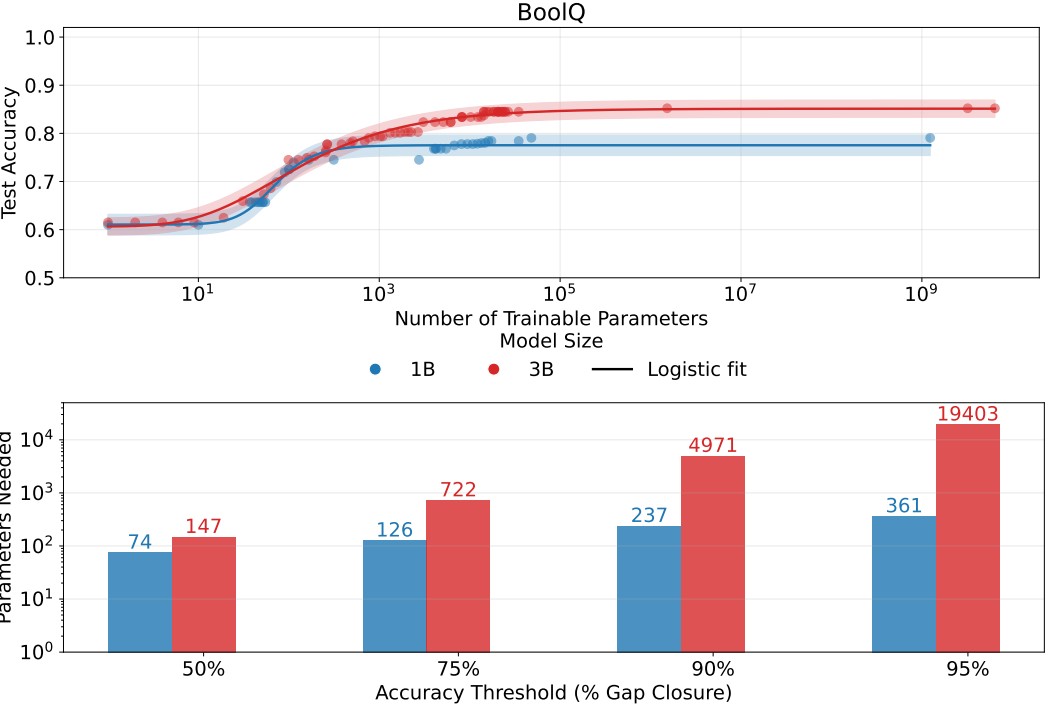

Figure 10: **Fine-tuning ~100 parameters in Llama-3.2-1B and Llama-3.2-3B recovers 50% of the performance gap on BoolQ.** $2\sigma$ standard errors are shown as shaded regions.

information Bob needs to know to replicate Alice's training on his end), update the model after each label, and code that label with its instantaneous cross-entropy loss. This "online" code length converges in expectation to the true MDL when the learner (i.e., model) is Bayes-optimal.

Perez et al. [18] introduce Rissanen Data Analysis to replace MDL with a block-wise code that retrains the model (calculating loss and updating gradients) only after a batch of examples has been seen, until all examples in the dataset have been seen a single time. This still constitutes an upper bound on the minimum information Alice must send because each batch is encoded with a less recent model than in the prequential setting.

Canonically, RDA either keeps the model architecture and optimizer fixed or charges $\log_2 M$ additional bits if an ensemble of $M$ learners is allowed, averaging codelengths over five random seeds. We adopt the same formulation.

We additionally ask the question, "How much do you have to specify to the model to get it to demonstrate knowledge it already has?". An intuitive way of measuring how much guiding a model needs to be able to access (some fraction of) a pre-existing capability is by counting the number of parameters that must be updated for it to be able to achieve a certain performance threshold on a task that requires the skill. In our elicitation framework, we employ MDL to measure how efficiently a model can encode task labels during its initial exposure to the training data. We compute MDL using the first epoch of training from a randomly initialized LoRA adapter, not from a converged model, as this provides a measure of how quickly a model can develop an efficient task representation given minimal parameter updates.

Comparing MDL and final test accuracy for a given model then provides information about how well a model can quickly develop a compact description of the task, as well as information about the model's ultimate asymptotic performance when given effectively unlimited parameter updates. For most tasks, MDL compression tracks accuracy improvements closely (Figure 5). However, divergences between MDL and accuracy curves (as seen for Llama-3.2-1B on ARC-Challenge) provide diagnostic information: when accuracy improves without commensurate MDL compression, the model may be learning rather than eliciting.

We can then recast our fine-tuning elicitation setting as a communication protocol between Alice and Bob:

All header terms are paid once, before the first label is transmitted, and the final summation term in the message is the usual prequential code computed only on the labels, as the inputs $x$ and backbone $M_0$ are shared *a priori*, following Perez et al. [18].

Our MDL calculation proceeds as follows:

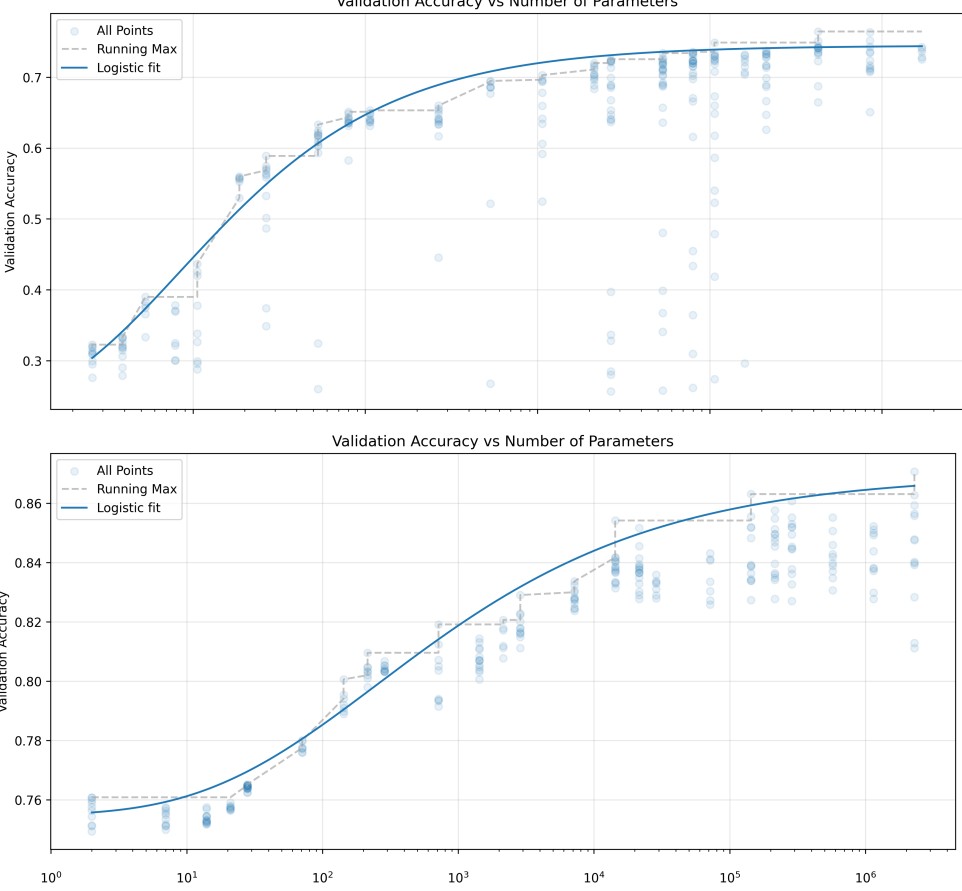

Figure 11: Validation accuracy for all hyperparameter configurations for Llama 3.2 1B and Llama 3.2 3B fine-tuned on ARC-Easy. Each semi-transparent point indicates a fine-tuned model with a unique combination of hyperparameters and random seed. The logistic fit to the Pareto frontier is denoted by a solid blue line, and a gray dashed line connects the points that define the Pareto frontier. For both models, nearly all 250+ hyperparameter and seed combinations (5 seeds with 50+ hyperparameter combinations each) for each parameter budget lie close to the Pareto frontier (within 2 percentage points).

1. **Initialization:** Given parameter budget $k$, randomly select $k$ parameters from LoRA adapters using seed $s$
2. **Hyperparameter selection:** Through Bayesian optimization, identify hyperparameters $h$ that maximize validation accuracy
3. **Prequential MDL computation:** For the optimal $(s, h)$ configuration, compute prequential loss during the first epoch only

Following the prequential MDL formulation [18, 32–34], we encode each example sequentially using the model trained only on previously observed examples:

$$\mathcal{L}_{\text{preq}} = -\log P(\mathcal{D}|\hat{\theta}) = -\sum_{t=1}^{T} \log(y_t|x_t; \theta_{t-1}), \tag{11}$$

in which $t$ indexes gradient steps in the first epoch, ensuring each encoded data point is unseen at encoding time. Practically, this is done by initializing the pretrained model, presenting each training example (or batch) exactly once, performing a gradient step immediately after an example (batch) has been seen, and accumulating the per-example cross-entropy losses over all examples (batches). This directly measures the incremental compression efficiency of a model during fine-tuning.

**Intuitive Interpretation:** Consider MDL as measuring how efficiently a model can "compress" its understanding of a task within its first pass over the training dataset. If updating merely 10 parameters enables the model to compress task labels by hundreds of bits within the first epoch alone, this suggests the model already possesses a

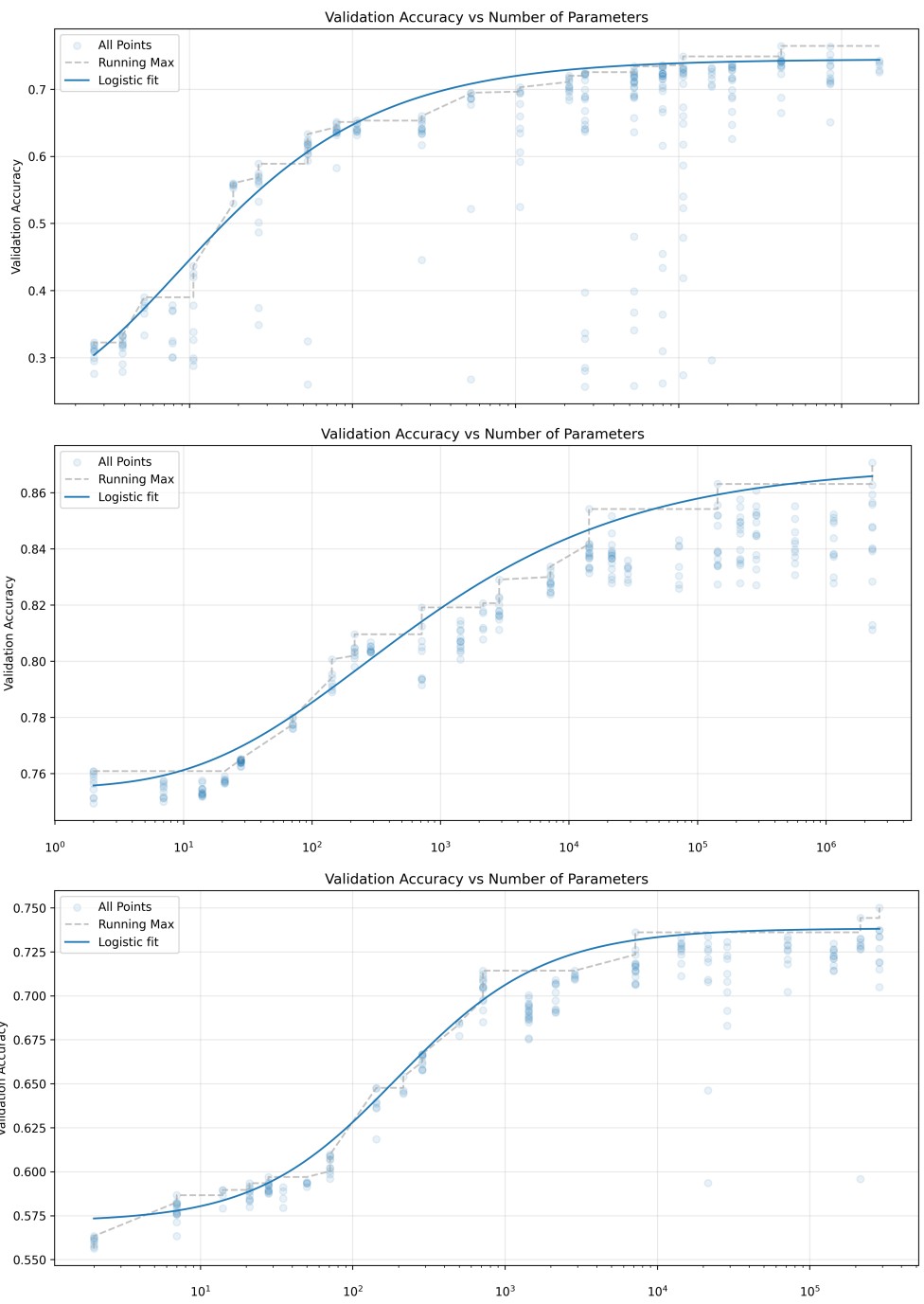

Figure 12: Llama-3.1-8B fine-tuned on ARC-Challenge. Each semi-transparent point indicates a fine-tuned model with a unique set of hyperparameters and random seed. The logistic fit to the Pareto frontier is denoted by a solid blue line, and a gray dashed line connects the points that define the Pareto frontier.

Table 3: **Casting PEFT elicitation as an Alice → Bob communication game.**

| Element | Description |
|---|---|
| Alice & Bob | Identical copies of pretrained base model $M_0$ |
| Shared knowledge | Base model $M_0$, learning algorithm $A$, dataset inputs $\{x_1, ..., x_n\}$ |
| Alice's goal | Communicate dataset labels $\{y_1, ..., y_n\}$ to Bob |
| Communication (message) | (i) A one-shot header describing number of trainable parameters $k$ to initialize the LoRA adapters $\theta_k$ with, parameter selection seed $s$, hyperparameters $h$; + (ii) block-wise stream of label codes |
| Instructions for accessing capability $f$ | The adapter update $\Delta\theta_k$ containing $k$ learned weights that Bob obtains as he trains his copy of $M_0$ |

structured representation of the task—the parameters merely serve as a "key" to unlock this latent knowledge. Conversely, if compression improves only marginally until thousands of parameters are modified, the capability may likely require learning from scratch rather than elicitation, as it cannot be easily unlocked or accessed.

---

**Algorithm 1** Prequential MDL Computation.

---

**Input:** Base model $M_0$, dataset $\mathcal{D} = \{x_i, y_i\}_i^N$, parameter budget $k$
Initialize model $M_0$ with LoRA adapters with $k$ trainable parameters, $\Delta\theta_k$, (using seed $s$)
Initialize MDL, $\mathcal{L}_k = 0$
**for** each batch $B$ in epoch 1 **do**
    Compute loss $\ell_B = \mathrm{CrossEntropy}(M_0(\Delta\theta_k, B), \mathrm{labels}(B))$
    $\mathcal{L}_k \mathrel{+}= \ell_B \times |B|$
    Update trainable parameters $\Delta\theta_k$ via gradient descent
**end for**
Return $\mathcal{L}_k$

---

**Interpretation for elicitation:** In scenarios with high compression and low trainable parameter counts, the model rapidly develops an efficient encoding scheme with a limited set of locations where it can make changes, suggesting it leverages pre-existing structured representations. In scenarios with low compression, the model struggles to compress the label stream efficiently, indicating that the capability is not easily accessible or utilized—potentially because it is absent. This differs from standard fine-tuning analysis because we measure compression during initial learning, not post-convergence, and explicitly account for the information cost of specifying which parameters to train. The resulting compression rate then reveals how much "latent structure" the model can immediately access.

The MDL compression $\Delta_k = \bar{\mathcal{L}}_p^{(0)} - \bar{\mathcal{L}}^{(k)}$ is therefore a direct, information-theoretic measure of how helpful the latent capability is when $k$ parameters are available to specify how to access it. In the language of Alice and Bob, if granting Alice access to $f$ shortens the MDL, then $f$ must convey information that was previously missing. The magnitude of the MDL drop, measured in bits, quantifies the intrinsic value of the capability to the task.

In our setting, every capability is presumed to be latent in the pretrained (base) model $M_0$. A small adapter $\Delta\theta_k$ with only $k$ trainable weights serves as a pointer to that capability; training enables the model to develop this pointer, as $\Delta\theta_k$ plays the role of invoking $f$. If a short message ($\Delta\theta_k$) lets the model compress the subsequent label stream by hundreds of bits, it is highly probable that the original base model already contained a highly structured representation of the task. Conversely, if the MDL shrinks slowly or not at all until $k$ becomes large, the capability was likely absent and must be learned rather than elicited.

# J  Data and Code Availability

GSM-8K-CoT-Choice dataset and evaluation scripts will be released with publication.

- LoRA implementation with random masking
- Bayesian optimization for ultra-low parameter regimes
- MDL computation utilities
- Frontier extraction and fitting procedures

