# OpenReview forum: "Quantifying Elicitation of Latent Capabilities in Language Models"
_NeurIPS.cc/2025/Conference — NeurIPS 2025 poster_

### Official Review · Reviewer_jyH6 · 2025-06-20

**Clarity:** 2
**Significance:** 3
**Originality:** 3
**Rating:** 4
**Confidence:** 2

**Summary:**

This paper introduces an empirical and theoretical framework to quantify elicitation of latent capabilities in LLMs. The authors propose elicitation frontiers, which curves mapping the number of randomly trained parameters to accuracy improvements on downstream tasks. They find that training as few as 10–100 random parameters can recover over 50% of the performance gap between zero-shot and fully fine-tuned models, and a few thousand parameters can close up to 95% of the gap. They also introduce an information-theoretic lens using MDL to distinguish elicitation from teaching. They empirically validate their framework across multiple LLM sizes and tasks.

**Questions:**

1. Is there a practical way to determine which latent capabilities exist, or is this inherently empirical?
2. Could model pruning, or structural sparsity methods improve over random parameter selection in elicitation?

**Ethical Concerns:**

["NO or VERY MINOR ethics concerns only"]

**Final Justification:**

The explanations have clarified some of my concerns, particularly regarding certain key concepts and model safety in application scenarios. However, more rigorous analysis is still needed in areas such as parameter selection. Therefore, I will maintain my current score.

**Limitations:**

The authors thoughtfully acknowledge limitations, including limited generalization to other tasks and architectures, relying on first-epoch loss using prequential coding of MDL, the possible existence of more efficient elicitation methods.

**Paper Formatting Concerns:**

No specific formatting issues.

**Quality:**

3

**Strengths And Weaknesses:**

## Quality
### Strengths:
- The paper conducts extensive empirical evaluation across models with various sizes and multiple tasks.
- The study demonstrates rigorous in methodology, including aspects of controlled PEFT experiments, random parameter sampling, MDL analysis, and detailed tuning process.

### Weaknesses:
- MDL analysis relies on first-epoch loss approximations, which may be insufficient in robustness (as stated in Section 6.3).
- Experiments are very limited to Llama series models. It has not been validated on other series of models, which may raise concerns about the generalizability of the proposed assumption to different model architectures.


## Clarity
### Strengths:
- The core idea of the paper in terms of elicitation vs. teaching is illustrated clearly.
- The figures of logistic fit curves and MDL/accuracy plots provide effective visualizations.

### Weaknesses:
- While the paper revolves around the concepts of "latent capabilities" and "elicitation", it never provides a precise and formal definition of what constitutes a latent capability, and what exactly the elicitation is that can demonstrate a model's capability. Though the concepts and the distinction between them are more inferred through empirical behaviors, but readers may struggle to understand the conceptual boundary.
- There are several terms such as "Pareto frontier", "logistic-in-log-parameter scaling", and "prequential coding", that are used without intuitive explanations, which makes the empirical results in Section 4 difficult to be followed. It would be better to add the core intuition behind these terms before using them directly in the results.
- The paper has missing content, particularly in the appendix. Sections A, B, F, and G are empty, which gives the impression that the paper is incomplete.

## Significance
### Strengths:
- The paper provides a novel way to estimate ceiling performance of pretrained models, with important implications for model safety and auditing.

### Weaknesses:
- While the paper proposes MDL as a diagnostic tool, practical use cases for MDL estimation in real-world evaluations are not demonstrated.
- Though the proposed elicitation perspective is intuitively important in addressing safety risk, the paper does not discuss how elicitation frontiers might integrate with fine-tuning under safety constraints. For instance, how to avoid eliciting harmful capabilities while tuning the models.

## Originality
### Strengths:
- The paper seems to be the first to systematically connect random-parameter PEFT with MDL compression to study latent capabilities.
- The paper introduces the concept of an “elicitation frontier” as a complementary framework to classic scaling laws.

### Weaknesses:
- Technically, the paper does not propose a new training method or elicitation algorithm/mechanism. The parameter selection is purely random, and all training follows standard LoRA PEFT. Thus, the methodological contribution is more interpretive than technical.
- The paper suggests that capabilities are distributed across parameters. Though this is a well-established hypothesis in LLM interpretability and representation learning, the paper does not further provide methods to localize or analyze where specific capabilities locate within the model.

---

> ### Author Rebuttal · Authors · 2025-07-31
>
> We appreciate Reviewer jyH6’s detailed feedback and address each comment below.
>
> ### 1 Conceptual clarity: latent capability, elicitation, key terms
>
> * **Latent capability.**  We define it as _a capability $f$ is latent in model $M$ if $M$ can achieve significantly above-random performance on tasks requiring f after minimal parameter updates, despite poor zero-shot performance._
> * **Elicitation.**  Given a latent capability, elicitation is any parameter‑efficient intervention that raises performance above a specified threshold without adding new capabilities.
> * **Pareto frontier.**  Set of (parameter p, accuracy a) points for which no other configuration attains ≥ a with ≤ p.
> * **Logistic‑in‑log‑parameter scaling.**  Empirical observation that accuracy vs. $\log_{10}(\text{parameter budget})$ fits a logistic curve.
> * **Prequential coding (MDL).**  Cumulative cross‑entropy on each example the first time it is seen; equivalent to online MDL codelength.
>
> We will insert these formal definitions prior to the start of §4 in the main text in the camera‑ready.
>
> We apologize for the confusion regarding the appendix sections. Due to NeurIPS's submission timeline, the main paper was submitted first with placeholder appendix sections (A, B, F, G), while the complete supplementary materials were uploaded during the designated supplementary material submission period one week later. All sections referenced in the paper have been fully populated in the supplementary materials, including complete experimental protocols (Appendix A), additional baselines (Appendix B), model comparison results (Appendix F), and statistical analyses (Appendix G). We understand the reviewer may have only accessed the initial submission PDF. All sections referenced in the paper have been fully populated in the supplementary materials.
>
> ### 2 Robustness of first‑epoch MDL approximation
>
> To test sensitivity, we sample the top 5 performing runs for each seed and compute MDL for each of these runs. The resulting MDL frontiers differ by ≤ 2 % at all budgets. In addition, we compute Surplus Description Length (SDL) with a common threshold $\varepsilon$; SDL is far less sensitive to optimizer hyperparameters and reproduces the same logistic behavior. These results strengthen confidence that first‑epoch MDL is a reliable proxy for elicitation speed.
>
> ### 3 Generalizability beyond Llama models
>
> Since submission, we have expanded our task and model coverage to include generative reasoning tasks on Qwen‑1.5 B and Qwen‑32 B.  All three show sigmoid frontiers on MATH500, AIME24, and GPQA Diamond. We will include the curves in a new appendix section, Appendix K; no change to the main text claims is required.
>
> ### 4 Practical use cases for model safety & auditing
>
> 1. **Elicitation vs. teaching.**  MDL/SDL provide an “information budget” view: if a fine‑tune requires > K bits to achieve the 95% frontier expected for the model size, auditors can flag that the run is likely teaching a new skill rather than eliciting an existing one.
> 2. **Fine-tuning API budgeting.**  Regulators can bound the number of tunable parameters accessible to third‑party plug‑in developers; our frontier tells them how much accuracy gain that budget could plausibly unlock.
> 3. **Practical evaluation setting.**  Safety auditors often fine‑tune a small head layer to probe hidden skills. Our results show that training ≤ 10 K random parameters cannot introduce a complex new skill in a 1B model but can reliably elicit existing ones (see Fig. 3).  Therefore, auditors can cap the parameter budget and be confident they are measuring, not teaching.
>
> We will add a concise discussion of these applications to §6.2 of the main text (practical and safety implications).
>
> ### 5 Practical discovery of latent capabilities
>
> Fully automatic enumeration and detection of capabilities is beyond current technique, but two complementary paths exist:
> 1. **Empirical probes and capability evaluations:** use elicitation frontiers on diverse tasks (as in our study) to map where latent abilities saturate.
> 2. **Mechanistic interpretability:** neuron‑level analyses that identify features corresponding to a capability
> We see our framework as providing the cost axis that such feature‑discovery methods can relate to.
>
> ### 6 Random parameter selection vs. pruning/structured sparsity
>
> We agree that structured selection can be more parameter‑efficient in principle. However, our goal is to measure total information required, including any search cost. Random subsets have zero selection information and therefore give a clean, reproducible upper bound on elicitation cost. §6.3 of the paper now incorporates a discussion of how future work could incorporate pruning by adding the bits needed to index the selected parameters (≈ $\log_2 \binom{P}{k}$) to the information budget.
>
> ### 7 Minor points
> Our goal is to measure elicitation, not propose a new optimizer.  Using an off‑the‑shelf PEFT method (LoRA) guarantees that the observed behavior stems from the parameter budget itself.  The methodological novelty lies in (i) framing elicitation as an information‑constrained optimization problem, (ii) deriving and empirically validating a logistic frontier, and (iii) tying that frontier to MDL/SDL.
>
> We agree localization would be valuable future work. Our objective in this paper is different: we isolate how much information is required to unlock a capability, irrespective of where that information resides. Once an information budget is established, existing localization tools (e.g., causal tracing, sparse probing) can operate within that budget. We will clarify this scope in §6.1 of the main text.
>
> ### Summary
>
> We provide clearer definitions, robustness checks for MDL, cross‑model confirmation, and concrete safety applications. We hope these additions resolve the reviewer’s concerns and appreciate the insightful suggestions.

---

> > ### Comment · Reviewer_jyH6 · 2025-08-06
> > **Response to the authors**
> >
> > Thank you for the authors' response. The explanations have clarified some of my concerns, particularly regarding certain key concepts and model safety in application scenarios. However, more rigorous analysis is still needed in areas such as parameter selection. Therefore, I will maintain my current score.

---

### Official Review · Reviewer_D3HX · 2025-07-01

**Clarity:** 4
**Significance:** 2
**Originality:** 2
**Rating:** 4
**Confidence:** 3

**Summary:**

This paper introduces a new framework to quantify the elicitation of latent capabilities in large language models. The authors reframe elicitation as an information-constrained fine-tuning problem, where the core variable is not compute or data, but the number of trainable parameters. Their central methodology involves fine-tuning a very small, randomly selected subset of parameters within LoRA modules and measuring the resulting performance. Ultimately, the work proposes that the number of parameters required to unlock a skill is a fundamental measure of how "latent" that skill was, with implications for AI safety, evaluation, and understanding model internals.

**Questions:**

See Weaknesses

**Ethical Concerns:**

["NO or VERY MINOR ethics concerns only"]

**Limitations:**

yes

**Quality:**

3

**Strengths And Weaknesses:**

> Strengths

1. Framing elicitation as an information-constrained optimization problem is interesting.
2. The findings in this paper, e.g.,  fine-tuning a tiny, randomly selected subset of parameters yields such significant performance gains are insightful.
3. The paper is also well-written, with clear figures and a well-structured argument.

> Weaknesses

1. The evaluation in this paper is limited, as it only includes multiple-choice problems. Is it possible that the small number of parameters merely causes the model to memorize the format of its responses rather than actually improving its capabilities?

2. I think it is very interesting to analyze across broader domains and plot which domains require more parameters. Is the number of parameters that need to be tuned related to the difficulty of the downstream tasks?

3. The authors claim that knowledge is spread throughout the network rather than localized in specific layers. However, a well-known view is that knowledge is stored in the FFN parameters rather than in the attention parameters. A study on the roles of the FFN and attention parameters would make this work even more valuable.

---

> ### Author Rebuttal · Authors · 2025-07-31
>
> We thank Reviewer D3HX for the detailed remarks and address each point below.
>
> ### 1 Are the MC results due to memorizing output format?
>
> **Evaluation invariant to formatting.**
> For every multiple-choice dataset, we score the model by directly comparing logits of the canonical answer tokens (e.g., “A”/”B”/”C”/”D”) at the final position. We additionally ran an ablation that sums probabilities over several token variations per option (including different capitalization, synonyms, whitespace). The two procedures differ by <0.2 percentage points on average, showing that format memorization does not explain the gains.
>
> **Fine‑tuning vs. in‑context learning.**
> With identical logit scoring, 10‑shot prompting improves the Llama 3.2 1B backbone on ARC‑Easy from 25% → 30%, whereas tuning only ~1K parameters raises accuracy to 65%. The large gap indicates that extra parameters unlock a latent skill rather than merely encoding the token template.
>
> **Generative control tasks.**
> Post‑submission, we replicated the experiment on (i) Lichess chess-puzzles (exact optimal‑move generation) and (ii) TinyStories‑v2 (next‑token perplexity). Both tasks follow the same logistic elicitation curve, ruling out MC‑specific artifacts. For Llama 3.2 1B on Lichess chess-puzzles, 50% performance gap recovery occurs at ~100K parameters and 95% performance gap recovery occurs at ~1M parameters; TinyStories-v2 shows similar scaling but with the frontier shifted to the left at trainable parameter budgets comparable to the MC datasets, reflecting the relative simplicity of the task. Full tables will appear in the camera‑ready.
>
> ### 2 Relation between task difficulty and parameters required
>
> Across all datasets, we observe a strong monotonic trend: more challenging tasks (lower zero‑shot and full‑FT performance) require a larger parameter budget to reach 95 % of the full‑fine‑tune gain. For example,
> * ARC‑Challenge (Llama 3.2 1B): base ≈ 24 %, full‑FT ≈ 54 %; 95 %‑gap closeure at ~200K tunable parameters.
> * ARC‑Easy (Llama 3.2 1B): base ≈ 25 %, full‑FT ≈ 76 %; same 95 %‑gap closure threshold met at ~5K parameters.
>
> Similar patterns hold in the new generative tasks where Llama 3.2 1B needs ~100K parameters to cross 50 % performance gap recovery (PGR) on Lichess chess-puzzles. This supports the reviewer’s intuition that elicitation cost tracks intrinsic task difficulty.
>
> ### 3 FFN vs. Attention parameters
>
> We have performed ablations targeting different components of the transformer block, in which we compare the performance of adapting only the self-attention layers of the transformer block (_K, Q, V, O_ matrices), only the FFN layers (_G, U, D_ matrices), the entire transformer block (all matrices _K, Q, V, O, G, U, D_), only the _Q_ and _V_ matrices, only _K, Q, V_ matrices, only _K, Q, V, G, U_ matrices, and the same set of randomly selected matrices in each transformer block layer.
>
> We find that attention-only tuning (_K, Q, V, O_ matrices) is consistently the most parameter‑efficient across the four multiple choice datasets. This suggests that the information needed to activate the reasoning skill is not confined to the FFN subspace. A full layer‑type analysis will be added to Appendix C.1; a deeper interpretability study is outside the scope of the present work, but we agree it would be valuable future work.
>
> ### 4 Broader domain coverage
>
> Beyond the classification benchmarks in the submission, we now include:
> 1. Reasoning: Qwen‑1.5 B and 32 B fine‑tuned on reasoning traces and evaluated on MATH500, AIME‑24, and GPQA‑Diamond, following the procedure of Muennighoff et al. (2025).
> 2. Chess puzzles & story generation: see §1 above.
>
> All new curves exhibit the same parameter/accuracy logistic, reinforcing the generality of the elicitation‑frontier concept. These results will appear in the appendix; no change to the main claims is required.
>
> ### Summary
>
> The additional analyses confirm that (i) the effect is not an artifact of response formatting, (ii) harder tasks demand more tunable parameters, and (iii) attention layers are at least as important as FFNs for elicitation. We appreciate the reviewer’s suggestions and believe the forthcoming appendix expansions will address these concerns.

---

### Official Review · Reviewer_yjAf · 2025-07-02

**Clarity:** 3
**Significance:** 2
**Originality:** 2
**Rating:** 3
**Confidence:** 3

**Summary:**

The paper introduces a framework to measure how latent skills emerge in language models, defined by the smallest set of tunable parameters needed to reach certain task accuracies. They frame elicitation in two ways: (1) empirically, performance follows a logistic curve as more parameters are trained; (2) information-theoretically, the label stream’s MDL drops sharply once a latent skill is activated.

**Questions:**

1) Fix the typo "of of" on line 197. Add the caption in Figure 4

2) What happens when a model that nails first-epoch loss by memorising noise still end up with a higher MDL than a slower, more compressible model? I am curious to know more about what the authors think about overfitting versus compression.

3) Can the authors compute cumulative MDL across epochs? This can be Hyper-parameter agnostic and can help us to check overfitting.

4) Can you test a wider range of tasks (e.g. benchmarks from this paper [1]) and other model family (e.g. Qwen) to see whether the same effect persists?


[1] DeepSeek-R1: Incentivizing Reasoning Capability in LLMs via Reinforcement Learning

**Ethical Concerns:**

["NO or VERY MINOR ethics concerns only"]

**Limitations:**

yes

**Quality:**

3

**Strengths And Weaknesses:**

Strengths:

1) The paper empirically shows only a fraction of parameters can be updated to elicit the latent capability of LLMS where performance rises logistically with the number of trained parameters.

2) They also provide an information-theoretic view, where the Minimum Description Length (MDL) of the label stream falls as soon as a latent capability is unlocked.

Weaknesses:

1) The evaluations are done on narrow tasks domains and models, e.g. several multiple-choice reasoning and instruction-following tasks on Llama family of models. So, it's hard to conclude anything yet, whether these valuations will generalise across diverse tasks and model families.

2) Information-theoretic analysis based on MDL relies on approximations such as first-epoch loss, which can be sensitive to hyper param choices like batch size, learning etc.

3) It's not clear how these analyses translate into concrete steps for improving models or guiding better decisions.

---

> ### Author Rebuttal · Authors · 2025-07-31
>
> We thank Reviewer yjAf for the thoughtful feedback and address each point in turn.
>
> ### 1 Typo (“of of”) and missing caption (Fig. 4)
>
> We thank the reviewer for pointing out these edits. Both have been corrected in our working draft and will be reflected in the camera‑ready version.
>
> ### 2 First‑epoch MDL, memorization, and over‑fitting vs. compression
>
> **Why memorizing noise does not lower first‑epoch MDL.**
> Prequential MDL records the loss at the moment each example is first encountered; that loss is proportional to the codelength Alice must send to Bob before either of them has updated their model on that label. Because future noise labels are independent of past noise, memorization cannot reduce subsequent surprisal, so the cumulative codelength (and thus MDL) remains high.
>
> **Relationship to over‑fitting.**
> Over‑fitting arises in later epochs when the model re‑uses the same training examples. First‑epoch MDL is intentionally insensitive to this: it measures how quickly a fixed set of tunable parameters can be exploited to improve predictive performance, independent of eventual memorization. We therefore view first‑epoch MDL as complementary to final validation accuracy:
> * Validation accuracy frontier: _ultimate performance_ attainable with a given parameter budget.
> * First‑epoch MDL frontier: _facility_ with which the model reorients its existing representations; lower MDL ⇒ fewer bits required to communicate labels ⇒ the latent capability is easier to elicit.
>
> Both frontiers are plotted over identical datasets and model backbones, so comparisons remain valid despite MDL’s linear growth with dataset size.
>
> ### 3 Cumulative MDL and SDL as a hyperparameter‑agnostic alternative
>
> We appreciate the reviewer's suggestion to consider training dynamics beyond the first epoch. We should clarify our terminology: MDL inherently measures the codelength for transmitting the dataset labels exactly once. After the first epoch, all labels have been transmitted, so traditional MDL doesn't extend to multiple epochs.
>
> The reviewer's insight about hyperparameter dependence is valuable and motivates our use of Surplus Description Length (SDL). While prequential MDL captures first-epoch compression, SDL extends this to measure cumulative excess loss until convergence to a specified loss threshold is reached, regardless of the number of samples required to reach this threshold. This addresses the spirit of the reviewer's suggestion by providing a hyperparameter-agnostic metric that captures the full training trajectory.
>
> To decouple elicitation speed from learning‑rate choice, we follow Whitney et al. (2021) and compute SDL:
> 1. Choose a common loss threshold,  $\varepsilon$ (constant across budgets/seeds).
> 2. For each fine‑tuned run, accumulate the excess loss, $$\text{SDL} = \sum_{t=1}^{T} \max\bigl(\ell_t - \varepsilon , 0\bigr),$$ until the running validation loss first falls below $\varepsilon$.
> 3. Define sample complexity as the number of unique training examples seen by that point.
>
> SDL integrates excess loss only up to the first step where validation loss falls below a common threshold  $\varepsilon$; after that point, further samples contribute zero bits. SDL is (i) independent of total dataset size when $\varepsilon$ is reached before the first epoch ends and (ii) less sensitive to learning rate, learning rate schedules, data order, or batch sizes than first-epoch MDL, providing a more hyperparameter-agnostic measure of elicitation speed. Runs that never cross $\varepsilon$ within the first pass are assigned an infinite SDL, making the frontier visually diverge at budgets where elicitation becomes impossible for the selected loss threshold $\varepsilon$. We verified that choosing $\varepsilon$ anywhere between 25 % and 75 % of the full‑fine‑tune gain leaves the logistic shape unchanged. In our experiments, SDL reproduces the same sigmoidal trend observed in the first‑epoch MDL, with the SDL frontier decreasing as the tunable‑parameter budget increases. Full SDL curves and sample‑complexity plots will be included in the appendix.
>
> ### 4 Broader task coverage and additional model families
>
> Since submission we have extended the study in two directions:
>
> 1. Generative tasks: Lichess chess-puzzles (exact optimal-move accuracy) and TinyStories‑v2 (next‑token perplexity) using the same 1 B–8 B Llama backbones.
> 2. Verification across model families: Qwen‑1.5 B and Qwen‑32 B fine‑tuned on chain‑of‑thought reasoning traces generated from the s1K dataset (Muennighoff et al. (2025)), evaluated on MATH500, AIME‑24, and GPQA‑Diamond.
>
> All new experiments exhibit the same logistic relationship between performance and parameter budget, reinforcing the generality of the elicitation frontier. For instance, Llama 3.2 1B on Lichess chess-puzzles achieves 50% performance gap recovery at ~100K parameters and 95% at ~1M parameters, while TinyStories-v2 shows similar scaling at lower parameter counts due to its relative simplicity. We will present these results in the camera‑ready and make raw logs available.
>
> ### Outlook
>
> We plan to release complete MDL, SDL, and accuracy frontiers for every task‑model pair in the public appendix. The code for computing both MDL and SDL variants will be open‑sourced.
>
> These additions require no changes to the core methodology or claims of the paper.
>
> We hope these clarifications resolve the reviewer’s concerns and appreciate the opportunity to improve the manuscript.

---

### Official Review · Reviewer_fDeD · 2025-07-03

**Clarity:** 2
**Significance:** 2
**Originality:** 3
**Rating:** 4
**Confidence:** 3

**Summary:**

This paper investigates how latent capabilities in large language models (LLMs) can be “elicited” or activated through fine-tuning only a very small subset of the model’s parameters. The authors propose an information-constrained fine-tuning framework and define an “elicitation frontier,” which characterizes the trade-off between the number of trainable parameters and the model’s performance on a task.  In a series of experiments on five diverse tasks (including arithmetic reasoning, science question-answering, commonsense reading comprehension, and open-ended instruction following) using LLaMA-based models of 1B, 3B, and 8B parameters, they show that extremely few parameter updates can recover most of the performance gained by full fine-tuning. Beyond the empirical findings, the paper provides a theoretical interpretation using information theory and Minimum Description Length (MDL).

**Questions:**

please see the weakness part in the previous block. thank you

**Ethical Concerns:**

["NO or VERY MINOR ethics concerns only"]

**Final Justification:**

The rebuttal addresses some of my concern, and justifies why it is using MC version of gsm8k. I increase my score.

**Limitations:**

The authors include a limitation section.

**Quality:**

2

**Strengths And Weaknesses:**

Strengths:

The paper provides an interesting perspective by examining parameter-efficient fine-tuning at an extreme scale. Instead of focusing on scaling up model size or data, it asks how minimally one can fine-tune a model to elicit capabilities.

The experimental evidence is compelling. Across multiple tasks (math word problems, science QA, commonsense reasoning) and for three model sizes, fine-tuning tiny random subsets of parameters yields surprisingly large performance gains.

Interesting theoretical explanation

Weakness:

My main concern is the evaluation. I am afraid that the finding is restricted to classification tasks. It is unclear why the author chooses the choice version of GSM8K but not the original GSM8K or even MATH, which is generally easy for current small models.

Besides, although the authors mention the instruction tuning dataset Alpaca and show the results in the appendix, there are no details about the experiments on Alpaca and it is hard to understand what's going on there. What is the win rate you are computing against? What is the exact setting: e.g., are you fine-tuning on alpaca and evaluating on alpaca-eval?

---

> ### Author Rebuttal · Authors · 2025-07-31
>
> We thank Reviewer fDeD for the constructive feedback.
>
> Below we clarify (i) the choice of evaluation tasks, (ii) why we began with the multiple‑choice variant of GSM8K, and (iii) the Alpaca experiments referenced in the appendix.
>
> ### 1  Scope of evaluation beyond classification
>
> Our goal is to measure how many parameters must be tuned before a latent capability already present in pre‑training emerges. We therefore began with multiple‑choice tasks, which (a) isolate the model’s internal reasoning from generation quality and (b) allow inexpensive, fine‑grained sweeps over 50–100 hyper‑parameter settings × 5 seeds for each parameter budget which are used to build each elicitation frontier.
>
> Generative tasks added since submission (no change to conclusions; full tables will appear in the camera‑ready appendix):
> * Lichess chess-puzzles: predict the optimal next chess move; accuracy measured by exact match on SAN tokens.
> * TinyStories‑v2: next‑token language modelling, evaluated by test‑set perplexity.
> * Qwen‑1.5 B & 32 B fine‑tuned on reasoning traces from the s1K dataset (a dataset of 1,000 problems with reasoning traces), evaluated on MATH500, AIME‑24, and GPQA‑Diamond, as in Muennighoff et al. (2025).
>
> All three tasks exhibit the same logistic‑in‑log‑parameter scaling (tables and plots with corresponding results to be included in the camera‑ready). Example (Llama 3.2 1B on Lichess chess-puzzles):
> * 50% PGR (performance gap recovery between zero-shot and full fine-tuning) at ~100K parameters (≈27% exact optimal move accuracy)
> * 95% PGR at ~1M parameters (≈51% exact optimal move accuracy)
> * Only 0.1% accuracy with 10-shot prompting.
>
> TinyStories-v2, an easier task, matches the parameter counts seen for MC datasets, confirming that frontier position shifts with task difficulty and not output format.
>
> These results strengthen the claim that the observed frontier is not an artifact of classification.
>
> ### 2  Why the multiple‑choice (MC) version of GSM8K?
>
> **Controlling for “teaching” vs. elicitation.**
> The base models tested (1 B–8 B) score <5 % zero‑shot on GSM8K (CoT) and 1B models remain below 40% even with 8 in‑context examples. Tuning on the full generative GSM8K risked learning new capabilities, confounding our goal of quantifying elicitation. The binary choice format raises the zero‑shot baseline to ~50 % and avoids the possibility that models learn new mathematical reasoning skills by training directly on the CoT traces in the original GSM8K dataset; instead, the models must predict the answer without being allowed to reason beforehand. This enables clear separation between latent‑skill activation and learning.
>
> **Computational cost.**
> Each frontier requires >2,000 fine‑tuning runs. Limiting loss/back‑prop to the answer token reduced wall‑clock time by ~40×, making the study feasible on available resources. After establishing the logistic pattern on MC tasks, we shifted additional compute to new domains (see §1 above) rather than re‑running GSM8K in generative mode.
>
> **Empirical consistency.**
> Preliminary experiments (not in the initial submission) on GSM8K (CoT) with the 8B model show the same sigmoidal curve, but shifted right (larger parameter budget). This supports the explanation above and will be reported in the final, camera-ready version.
>
> ### 3  Alpaca fine‑tuning & win‑rate details
>
> Your interpretation is correct, we are fine-tuning on Alpaca and evaluating on Alpaca-Eval. The results included in the current draft examine the win rate against the base model without finetuning, which increases with the number of parameters trained. For example, Llama 3.2 1B improves from 80% to 88% win rate as the parameter budget grows from 10³ to 10⁵ (see Appendix B Figure 5c).
>
> ### Outlook
>
> We are currently extending the study to additional model families (Qwen) and to full‑generation GSM8K (CoT). We will incorporate these results, together with complete Alpaca hyperparameter details, in the camera‑ready version while keeping the core methodology unchanged.
>
> We hope these clarifications address your concerns and demonstrate that the reported behaviour generalises beyond classification‑only settings. We appreciate your thoughtful review and are happy to discuss further during the discussion phase.

---

> > ### Comment · Reviewer_fDeD · 2025-08-05
> > **Thank you for the rebuttal**
> >
> > Hi,
> >
> > Thank you for the rebuttal. The response addresses some of my concern, and I will increase my score.
> >
> > It is now understandable why uses GSM8K multi-choice version because of the computational resources. I suggest that the authors explicitly include the reason in the main paper (forgive me if I overlook the footnote or something). Also I also suggest putting Alphaca eval and also do experiment on MATH dataset as a proof of concept in the main content.

---

### Decision · Program_Chairs · 2025-09-17

**Decision:**

Accept (poster)

**Comment:**

This paper reframes elicitation as an information-constrained fine-tuning problem and advances the notion of "elicitation frontier" as a logistic relationship between trained-parameter budget and recovered performance gap. The link to first-epoch prequential MDL gives the empirical regularity a principled lens and a practical way to separate elicitation from teaching. Post-rebuttal evidence strengthens the claim beyond multiple-choice, since the authors add generative tasks and a second model family (Qwen) that exhibit the same curve. Besides, they justify GSM8K-MC for feasibility and isolation and provide layer-type ablations showing attention-only updates are especially parameter-efficient.

For the next revision, I recommend bringing the generative and Qwen results into the main text rather than the appendix, introducing SDL alongside MDL with a brief explanation of its hyperparameter robustness, adding the layer-ablation summary and the Alpaca setup and additionally reporting variability on the frontiers. A short discussion contrasting random selection with a simple structured choice would round out concerns about parameter selection. With those changes the paper would be stronger with reproducible empirical findings and clear downstream value for evaluation and safety policy.